# Frame-wise Conditioning Adaptation for Fine-Tuning Diffusion Models in Text-to-Video Prediction

**Zheyuan Liu**                                                    *zheyuan.liu@adelaide.edu.au*
*Center for Augmented Reasoning, Australian Institute for Machine Learning, University of Adelaide*

**Junyan Wang**                                                    *junyan.wang@adelaide.edu.au*
*Center for Augmented Reasoning, Australian Institute for Machine Learning, University of Adelaide*

**Zicheng Duan**                                                   *zicheng.duan@adelaide.edu.au*
*Center for Augmented Reasoning, Australian Institute for Machine Learning, University of Adelaide*

**Cristian Rodriguez-Opazo**                          *cristian.rodriguezopazo@adelaide.edu.au*
*Center for Augmented Reasoning, Australian Institute for Machine Learning, University of Adelaide*

**Anton van den Hengel**                                  *anton.vandenhengel@adelaide.edu.au*
*Center for Augmented Reasoning, Australian Institute for Machine Learning, University of Adelaide*

**Reviewed on OpenReview:** *https://openreview.net/forum?id=HSAjl4LUHK*

## Abstract

Text-video prediction (TVP) is a downstream video generation task that requires a model to produce subsequent video frames given a series of initial video frames and text describing the required motion. In practice TVP methods focus on a particular category of videos depicting manipulations of objects carried out by human beings or robot arms. Previous methods adapt models pre-trained on text-to-image tasks, and thus tend to generate video that lacks the required continuity. A natural progression would be to leverage more recent pre-trained text-to-video (T2V) models. This approach is rendered more challenging by the fact that the most common fine-tuning technique, low-rank adaptation (LoRA), yields undesirable results. In this work, we propose an adaptation-based strategy we label Frame-wise Conditioning Adaptation (FCA). Within the module, we devise a sub-module that produces frame-wise text embeddings from the input text, which acts as an additional text condition to aid generation. We use FCA to fine-tune the T2V model, which incorporates the initial frame(s) as an extra condition. We compare and discuss the more effective strategy for injecting such embeddings into the T2V model. We conduct extensive ablation studies on our design choices with quantitative and qualitative performance analysis. Our approach establishes a new baseline for the task of TVP. Our code is open-source at https://github.com/Cuberick-Orion/FCA.

## 1 Introduction

Text-video prediction (TVP) (Gu et al., 2023) takes as input an initial frame or frames, and text describing a particular motion (Figure 1). On this basis it generates subsequent frames wherein the existing content undergoes the described motion. As a downstream video generation task, its configuration enables the study of instruction adherence, as well as temporal consistency in the context of video extension given conditioning frames, particularly in a fine-tuning setup. The established benchmark and datasets emphasize motions carried out by human beings or robot arms in manipulating objects.

Given the relatively limited training data for TVP, methods for this task typically involve fine-tuning a pre-trained generative model. The challenge, therefore, lies in guiding the pre-trained model in comprehending

and adhering to the multi-modal conditions. While current methods (e.g., (Gu et al., 2023)) have made admirable progress, they still fall short in generation quality, partly because they build upon text-to-image models (Rombach et al., 2022). Given the recent success of large-scale pre-trained text-to-video generative models (Hong et al., 2022; Yang et al., 2024b; Kong et al., 2024) particularly based on diffusion transformers (DiT) (Peebles & Xie, 2023), we propose to adapt such models towards the task of TVP. However, we find this goal non-trivial. The reasons are twofold. First, we empirically find that low-rank adaptation (LoRA) (Hu et al., 2022a), the standard method for fine-tuning generative models on domain-specific data, yields undesired results on TVP. Second, text-to-video models are pre-trained to align to a text prompt alone, whereas the task of TVP involves conditioning on the initial frames as well. Specifically, it requires the model to reason over the input frames and extend its future predictions upon them, while jointly taking the text prompt into consideration.

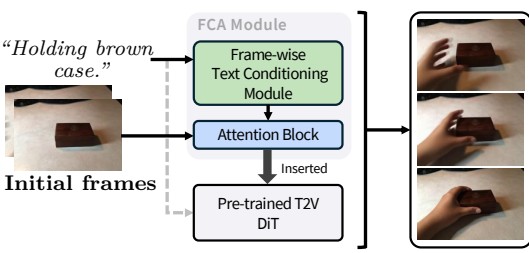

Figure 1: **Overview.** Text-video prediction (TVP) models generate subsequent video frames on the basis of the initial frame(s) and a natural language description of the required motion. Our method leverages a pre-trained text-to-video (T2V) diffusion transformer (DiT) model, while introducing an effective adaptation method (FCA) for fine-tuning. Our method integrates the initial frames, as well as frame-wise text conditions to aid the generation.

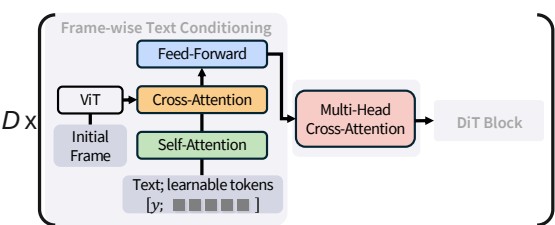

Figure 2: **Details of the frame-wise text conditioning module inspired by Q-Former (Li et al., 2023a), and its integration with FCA.** We only show one DiT layer here for clarity, but note that we separately apply a frame-wise text conditioning module to every layer. In total, we initialize and train $D$ such modules for the $D$ layers of the DiT. This figure complements Figure 3 (FCA module, bottom-left block). ViT stands for the Vision Transformer (Dosovitskiy et al., 2020).

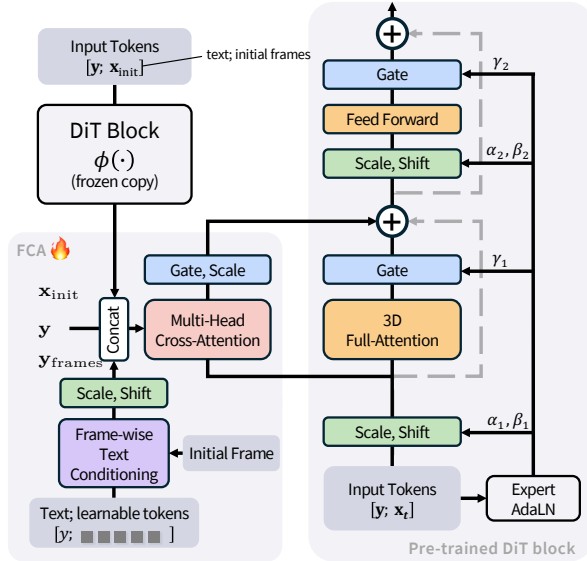

Figure 3: **An illustration of Frame-wise Conditioning Adaptation (FCA)** on diffusion transformer (DiT). **Right:** an arbitrary pre-trained DiT block (Peebles & Xie, 2023). **Left:** the proposed FCA module, introduced in Section 3.2, which incorporates frame-wise text conditioning discussed in Section 3.3. We only show one DiT layer here for clarity, but note that we separately apply the same modules to every layer. $\mathbf{y}$, $\mathbf{x}_t$ denote the text tokens and noisy latent for a DiT, respectively; $[\cdot ; \cdot]$ denotes concatenation. $\mathbf{x}_{\text{init}}$ represents the latent of the initial frames introduced in Section 3.2, and $\mathbf{y}_{\text{frames}}$ is the frame-wise text conditioning embeddings. A frame-wise attention mask is applied within the multi-head cross-attention module, which is omitted in this figure, see Section 3.3. $\alpha, \beta, \gamma$ are the learned parameters of the adaptive Layernorm (adaLN) (Peebles & Xie, 2023). The trainable module is marked with a fire symbol. See Figure 2 for details of the frame-wise text conditioning module.

The proposed approach requires an adaptation strategy tailored to the task setup. We have thus developed an effective design based on a DiT model, as shown in Figure 1. Specifically, we inject parallel attention modules

into the layers of the DiT, whose output is added back in a residual manner. Meanwhile, we freeze the pre-trained DiT to avoid disrupting its ability to process video. To integrate the initial frames as a condition, we propose to use a frozen copy of the DiT model to extract the latent of such frames, and incorporate this information into video generation through cross-attention. The strategy is similar to some zero-shot subject-driven personalized image generation methods (Jia et al., 2023; Purushwalkam et al., 2024; Duan et al., 2024) in that their approach is to capture the visual characteristics of the subjects from the user-provided images. We qualitatively validate that our design yields superior performance over standard fine-tuning techniques such as LoRA; and demonstrate the effectiveness of our approach in integrating the initial frames in comparison to well-established practices of enabling additional control, such as ControlNet (Hu & Xu, 2023). We report empirical findings and training tricks to aid future work in fine-tuning such DiT models via adaptation.

We additionally seek to apply frame-wise text conditioning to our method, which has been empirically demonstrated beneficial for TVP (Gu et al., 2023). The reason is thought to be that the raw text conditions in the TVP datasets (e.g., "pick something up" or "move away from the camera") are not specific enough to specify the intermediate stages of carrying out a motion. Without collecting extra annotations, we develop learnable modules that infer such frame-wise conditions through fine-tuning. The learned conditioning embeddings are inserted back to the parallel attention layers we introduced, where a frame-wise attention mask is applied. We experiment with various designs and share our insights on how to construct and apply such conditions efficiently. Collectively, our method, termed Frame-wise Conditioning Adaptation (FCA), surpasses existing work on TVP, thereby establishing a strong baseline for future research.

To summarize, we propose to apply recent large-scale pre-trained text-to-video generative models to the task of TVP. To accommodate the task requirements while maintaining a reasonable training cost, we introduce an effective adaptation-based fine-tuning strategy that incorporates frame-wise text conditions via learned modules to assist the learning. Our design additionally integrates the initial frames into the condition. We conduct extensive ablation studies to validate our method, while documenting empirical findings and training tricks for future research. Our method, termed FCA, improves on previous work both quantitatively and qualitatively on standard benchmark datasets. Specifically, we achieve a 40% reduction (relative) in the FVD metric on both Something-Something-V2 (Goyal et al., 2017) and EpicKitchen-100 (Damen et al., 2020) and an impressive 60% FVD reduction (relative) on BridgeData (Ebert et al., 2021).

## 2 Related Work

**Pre-trained Video Generation Model.** The impressive performance of diffusion models (Ho et al., 2020; Song et al., 2020b) has spurred a recent trend in developing large-scale pre-trained video generative models (Ho et al., 2022; Hong et al., 2022), which are viewed as a natural progression from the image generative task (Ramesh et al., 2022; Saharia et al., 2022; Rombach et al., 2022). Text is most commonly used as the conditioning signal for generating videos, i.e., text-to-video (T2V), analogous to what is seen in text-to-image (T2I). Architecture-wise, pioneer work (Singer et al., 2022; Wu et al., 2023b; Blattmann et al., 2023; Wang et al., 2023a; Zheng et al., 2024) often adopts the U-Net architecture (Ronneberger et al., 2015) and injects temporal modules alongside the existing spatial modules, thereby inflating an image generation model for video generation. A line of recent work (Yang et al., 2024b; Kong et al., 2024) adopts the diffusion transformer (DiT) (Peebles & Xie, 2023) with better scalability. In this work, we adopt CogVideoX (Yang et al., 2024b) for our base model, which utilizes 3D full attention that alleviates the need for separate spatial and temporal attention modules.

Although T2V is the most common paradigm for pre-trained video generation models, several recent work (Yang et al., 2024b; Kong et al., 2024; Blattmann et al., 2023) have also included a variant of their proposed T2V base model, that additionally supports an image for conditioning as the first frame. To adapt a pre-trained T2V model into image-to-video (I2V), the common practice (Girdhar et al., 2024) is to concatenate the extracted representation of the initial frame to the input noisy latent along the channel dimension, and perform further training. We note that the task of text-video prediction (TVP) is similar to I2V regarding the setup of input conditions. However, we opt for developing based on a T2V model, we refer readers to Section 4.1 for discussions.

**Incorporating Extra Conditions.** Multiple works seek to inject extra conditions to guide the generation. Various categories of conditions exist for different applications, including edges or depth maps (Guo et al., 2024; Chen et al., 2023), optical flow (Zhang et al., 2024; Liang et al., 2024), camera movements (Yang et al., 2024a) and pose information (Hu, 2024; Ma et al., 2024; Xu et al., 2024). Another video can also serve as guidance for video-to-video generation (Hu & Xu, 2023; Zhao et al., 2024; Wu et al., 2023c). Notably, a line of work exists that addresses the concept of "motion" (Shen et al., 2024; Shi et al., 2024; Zhang et al., 2024; Li et al., 2023b; Hu & Xu, 2023; Zhao et al., 2024; Dai et al., 2023), however, they either focus on the movement of a rigid body (e.g., a car moving) or a human being (e.g., riding a bicycle). As such, they are different from the concept of motion seen in the task of text-video prediction (TVP), i.e., the manipulation of objects. We note that different categories of conditions often require specific designs to inject into the generative model, as they differ in format and semantics.

**Text-Video Prediction (TVP).** TVP conditions the generation on both the initial frames and the text, which differs from the conventional task of video prediction (Feichtenhofer et al., 2016) that conditions solely on the previous frames (Ye & Bilodeau, 2024; Höppe et al., 2022; Voleti et al., 2022; Harvey et al., 2022). It is recently proposed by Gu et al. (2023) alongside a method termed Seer, where a pre-trained, U-Net (Ronneberger et al., 2015) T2I model is adapted for producing videos by inserting additional temporal attention layers, as discussed above. The authors discovered that frame-wise text conditions benefit the task, and introduced a text decomposition module built into the proposed architecture. In this work, we further advance the task of TVP by leveraging more suitable, video-generative pre-trained models, while also incorporating frame-wise conditions but with a tailored design to fit the architecture.

## 3 Method

The task of text-video prediction (TVP) takes in as input the initial $k$ frame(s) of a video $\{f_1, \cdots, f_k\}$ alongside the conditional text $y$ and aims at producing the subsequent frames $\{f_{k+1}, \cdots, f_n\}$. We note the entire video as the collection of all frames $F = \{f_i\}_{i=1}^n$. In this work, we leverage a pre-trained text-to-video (T2V) diffusion transformer (DiT) (Peebles & Xie, 2023) as our base model, where we propose our Frame-wise Conditioning Adaptation (FCA) design. We note the number of transformer layers in a DiT as $D$, as seen in Figure 2.

### 3.1 Preliminaries

**Denoising Diffusion Probabilistic Models.** Diffusion models (Ho et al., 2020; Song et al., 2020a; Dhariwal & Nichol, 2021; Song et al., 2020b) are a type of generative model that performs bi-direction operations. The forward direction takes in a sample $\mathbf{x}_0$ and iteratively adds Gaussian noise to it until a maximum step $T$ is reached. Opposite to it, the generative process starts with the Gaussian noise $\mathbf{x}_T$, where the model gradually estimates the noise to be subtracted, and ultimately produces the sample $\mathbf{x}_0$. The training objective of the model is to predict the to-be-subtracted noise:

$$L = \mathbb{E}_{\mathbf{x}_0, \mathbf{c}, \boldsymbol{\epsilon} \sim \mathcal{N}(\mathbf{0}, \boldsymbol{I}), t} \| \boldsymbol{\epsilon} - \boldsymbol{\epsilon}_\theta (\mathbf{x}_t, \mathbf{c}, t) \|^2, \tag{1}$$

where the diffusion model is denoted as $\boldsymbol{\epsilon}_\theta$ parameterized by $\theta$, $t \in [0, T]$ represents the time step of the process, and $\mathbf{x}_t = \alpha_t \mathbf{x}_0 + \sigma_t \boldsymbol{\epsilon}$ is the noisy data at step-$t$ with $\alpha_t, \sigma_t$ being functions of $t$. Here, $\mathbf{c}$ represents various conditions that can guide the generation.

Once trained, the diffusion model can be used for predicting samples from the Gaussian noise. Pioneer work (Dhariwal & Nichol, 2021) adopts classifier guidance that uses the gradients of a trained image classifier to control the generated samples. Subsequently, classifier-free guidance (Ho & Salimans, 2021) eliminates the need for a separate classifier model. Its sampling stage involves a linear combination of conditional and unconditional predictions, where the latter term replaces the condition $\mathbf{c}$ with an empty embedding $\varnothing$, as:

$$\tilde{\boldsymbol{\epsilon}}_\theta(\mathbf{x}_t, \mathbf{c}, t) = \lambda \boldsymbol{\epsilon}_\theta(\mathbf{x}_t, \mathbf{c}, t) + (1 - \lambda) \boldsymbol{\epsilon}_\theta(\mathbf{x}_t, \varnothing, t), \tag{2}$$

with $\lambda$ being the guidance scale. A noticeable characteristic of the diffusion model is the disparity between its training and inference operations. Specifically, one can apply one of many samplers for generating samples.

For instance, samplers such as DDIM (Song et al., 2020a) enable a faster generation by requiring only a small number of steps.

Recent work often adopts the latent diffusion model (LDM) (Rombach et al., 2022), which operates on the latent space as opposed to the raw pixel space, and therefore requires less computational resources. The encoding and decoding operation is through a frozen variational autoencoder (VAE). We omit the VAE in the notations for brevity and directly view $\mathbf{x}_t$ as the latent embedding below.

**Diffusion Transformer.**  While conventional diffusion models often adopt the U-Net (Ronneberger et al., 2015) architecture, recent variants leveraging diffusion transformer (DiT) (Peebles & Xie, 2023) have increasingly gained popularity thanks to its scalability. The DiT replaces the convolution layers of the U-Net with multiple layers of transformer modules, where each layer performs multi-head attention (Vaswani et al., 2017) over a sequence of tokens.

The sequence of tokens is effectively a concatenation between the input noisy latent and the conditioning embeddings, such as text embeddings $\mathbf{y}$ obtained through a pre-trained text encoder. Here, the input noisy latent of an image is patchified along the spatial dimensions to form a sequence of tokens. In the case of video generative models, multiple frames are treated as consecutive images to form a longer sequence. Positional embeddings (Dosovitskiy et al., 2020; Su et al., 2024) are often injected, similar to what is seen in the Vision Transformer (ViT) (Dosovitskiy et al., 2020). The different types of tokens separately go through adaptive Layernorm (adaLN) (Peebles & Xie, 2023), which involves scaling and shifting the embeddings to bridge the modality gap. Without harming clarity, we omit the patchifying and adaLN operations, and denote the token sequence received by each layer of the transformer as $\mathbf{Z} = \text{concat}\,[\mathbf{y}; \mathbf{x}_t]$.

Within each transformer layer of the DiT, the concatenated token sequence $\mathbf{Z}$ performs self-attention (Vaswani et al., 2017), as:

$$\mathbf{Z}' = \text{Attention}\,(\mathbf{Q}, \mathbf{K}, \mathbf{V}) = \text{Softmax}\left(\frac{\mathbf{Q}\mathbf{K}^\top}{\sqrt{d}}\right)\mathbf{V}, \tag{3}$$

where $\mathbf{Q} = \mathbf{Z}\mathbf{W}_q$, $\mathbf{K} = \mathbf{Z}\mathbf{W}_k$, $\mathbf{V} = \mathbf{Z}\mathbf{W}_v$ are the query, key, and value matrices computed by multiplying the token sequence $\mathbf{Z}$ with separate learnable weight matrices. Here, $\mathbf{Z}$ serves as the token sequence for query, key, and value simultaneously. The output $\mathbf{Z}'$ is then added to $\mathbf{Z}$ in a residual manner (He et al., 2016), followed by a feed-forward layer.

## 3.2 Adapting Pre-trained T2V Model for TVP

In this work, we adopt an effective adaptation-based fine-tuning strategy for DiT-based T2V model. As shown in Figure 3, within each layer of the transformer architecture, we insert an attention block parallel to the pre-trained attention block, whose output is added back to the pre-trained branch. In practice, we find it crucial to initialize a small learnable scaling factor (e.g., 0.1) and multiply it with the attention output. We freeze the pre-trained branch to best preserve its learned knowledge while training the inserted modules.

Unlike in the pre-trained branch where each attention block performs self-attention over the concatenated text and video tokens ($\mathbf{Z} = \text{concat}[\mathbf{y}; \mathbf{x}_t]$), in the inserted attention blocks, we modify the key and value token sequences to perform a cross-attention. Since the query sequence remains as $\mathbf{Z}$, the dimensionality of the output sequence is unchanged and can be added back. Specifically, we construct the key and value token sequences as the concatenation among the text token $\mathbf{y}$, the initial frames latent $\hat{\phi}(\mathbf{x}_{\text{init}})$ (see below), and the frame-wise text conditions $\mathbf{y}_{\text{frames}}$ (see Section 3.3).

**Integrating the Initial Frames.**  Since TVP requires the predicted frames to extend on the initial frames, it is necessary for us to inject the initial frames as conditions to the T2V model. As illustrated in Figure 3 (top-left), we append the latent embeddings of the initial frames to the key and value token sequences of the FCA attention layers. Consequently, the model learns to attend to the information within such embeddings through fine-tuning. Inspired by recent work on personalized image generation (Jia et al., 2023; Purushwalkam et al., 2024), we obtain such embeddings for each FCA layer using a frozen copy of the pre-

trained DiT model $\hat{\phi}(\cdot)$, where we feed the latent of the initial frames $\mathbf{x}_{\text{init}}$ as input and perform a forward pass, thereby extracting its intermediate layer-wise embeddings, noted as $\mathbf{X}_{\text{init}} = \left\{ \hat{\phi}(\mathbf{x}_{\text{init}})^i \right\}_{i=1}^{D}$.

To obtain $\mathbf{x}_{\text{init}}$, we treat the initial frames as a short video of only one or two frames, and encode it via the VAE (Yu et al., 2023a). As the DiT model is trained to receive a noisy latent, rather than a noise-free latent as input, we follow Duan et al. (2024) by adding a small noise (one step) to $\mathbf{x}_{\text{init}}$ before inputting it into the DiT. This ensures that the latent embedding adheres to the expected input data distribution while introducing minimum loss to its information.

## 3.3 Frame-wise Text Condition

We explore the use of frame-wise text conditions, which previous work (Gu et al., 2023) has demonstrated to be beneficial, as the text conditions in text-video prediction (TVP) datasets (e.g., "taking remote out of pen stand") are often too concise to depict the intermediate stages of carrying out a motion.

As in the previous work (Gu et al., 2023), without collecting additional text annotations, we seek to infer frame-wise text embeddings and use them to complement the vanilla text $y$ during generation. In essence, given the text tokens $\mathbf{y} \in \mathbb{R}^{l_t \times d}$, we aim to derive $\mathbf{y}_{\text{frames}} \in \mathbb{R}^{f \times l_t \times d}$ such that $f$, i.e., its frame dimension corresponds to that of the video tokens $\mathbf{x}_t$. Here, $l_t$ represents the length of the text token coming out of the text encoder while $d$ is the channel dimension; we omit the batch dimension for brevity.

**Frame-wise Text Conditioning Module Design.** Our frame-wise text conditioning module is inspired by Q-Former in BLIP-2 (Li et al., 2023a). As shown in Figure 3 (bottom-left), we initialize a sequence of learnable tokens for $\mathbf{y}_{\text{frames}}$, which are concatenated with the tokens of text $y$ obtained via tokenization and projection. Figure 2 illustrates the detailed architecture. Specifically, the two perform self-attention first, followed by a cross-attention layer where we introduce the initial frame $f_1$. $\mathbf{y}_{\text{frames}}$ is obtained following a final feed-forward layer.

The frame-wise text conditioning module is trained alongside the FCA attention layers. We find it ideal to initialize a separate such module per attention layer, instead of employing one module and feeding its output to all attention blocks of FCA (Figure 2). Our intuition is that every DiT layer learns different semantics; therefore, so should the conditioning embeddings. We denote the $i$-th layer frame-wise text conditioning module as $\boldsymbol{\gamma}^i(\cdot)$, therefore, $\mathbf{y}_{\text{frames}}^i = \boldsymbol{\gamma}^i(y, f_1)$.

**Applying Frame-wise Text Conditioning.** Given $\mathbf{y}_{\text{frames}}^i$ that corresponds to the $i$-th FCA attention layer, we append it to the key and value token sequences to perform a cross-attention, analogous to our approach of integrating the initial frames in Section 3.2. We learn a pre-normalization to $\mathbf{y}_{\text{frames}}^i$ mimicking the adaptive Layernorm (adaLN) (Peebles & Xie, 2023) seen in DiT, which is empirically beneficial.

To guide the frame-wise text conditioning modules to learn $\mathbf{y}_{\text{frames}}^i$ that serves as frame-wise condition, we construct an identity matrix for the attention mask on the frame dimension between $\mathbf{y}_{\text{frames}}^i$ and the video tokens $\mathbf{x}_t$. Recall that $\mathbf{y}_{\text{frames}}^i$ and $\mathbf{x}_t$ share the same length in frame. Effectively, the mask prevents video tokens of frame-$p$ from attending to text conditions of frame-$q$ when $p \neq q$. Meanwhile, we still include the vanilla text condition $\mathbf{y}$ in the cross-attention, and allow all video tokens to attend to it.

## 3.4 Training and Inference

During training, we follow the standard diffusion objective in Equation 1 to optimize the inserted FCA blocks. We incorporate the FCA attention layers into $\boldsymbol{\epsilon}_\theta$ parameterized by $\theta$, while separately denoting the frame-wise text conditioning modules as $\boldsymbol{\gamma}(\cdot)$. We freeze other parameters of the pre-trained DiT. The training objective is formulated as:

$$L = \mathbb{E}_{\mathbf{x}_0, \mathbf{X}_{\text{init}}, \mathbf{y}, \boldsymbol{\epsilon}, t} \| \boldsymbol{\epsilon} - \boldsymbol{\epsilon}_\theta \left( \mathbf{x}_t, \mathbf{X}_{\text{init}}, \mathbf{y}, \boldsymbol{\gamma}(y, f_1), t \right) \|^2, \tag{4}$$

where $\mathbf{y}$ denotes the conditioning text embeddings, and $\mathbf{X}_{\text{init}}$ represents the extracted latent of the initial frames. We perform no dropouts to the initial frames or text as we find it not beneficial for fine-tuning.

| | Methods | Pre-trained with | Text Condition | SSv2 | | BridgeData | | Epic100 | |
|---|---|---|---|---|---|---|---|---|---|
| | | | | FVD ↓ | KVD ↓ | FVD ↓ | KVD ↓ | FVD ↓ | KVD ↓ |
| 1 | TATS (Ge et al., 2022) | video | no | 428.1 | 2177 | 1253 | 6213 | 920.0 | 5065 |
| 2 | MCVD (Voleti et al., 2022) | — | no | 1407 | 3.80 | 1427 | 2.50 | 4804 | 5.17 |
| 3 | SimVP (Gao et al., 2022) | — | no | 537.2 | 0.61 | 681.6 | 0.73 | 1991 | 1.34 |
| 4 | MAGE (Hu et al., 2022b) | video | yes | 1201.8 | 1.64 | 2605 | 3.19 | 1358 | 1.61 |
| 5 | PVDM (Yu et al., 2023b) | — | no | 502.4 | 61.08 | 490.4 | 122.4 | 482.3 | 104.8 |
| 6 | VideoFusion (Luo et al., 2023) | text-video | yes | 163.2 | 0.20 | 501.2 | 1.45 | 349.9 | 1.79 |
| 7 | Tune-A-Video (Wu et al., 2023a) | text-image | yes | 291.4 | 0.91 | 515.7 | 2.01 | 365.0 | 1.98 |
| 8 | Seer (Gu et al., 2023) | text-image | yes | 112.9 | 0.12 | 246.3 | 0.55 | 271.4 | 1.40 |
| 9 | FCA (Ours) | text-video | yes | **67.7** | **-0.18** | **96.5** | **0.43** | **164.6** | **0.45** |

Table 1: **Quantitative results of video generation.** We report on Something-Something-V2 (SSv2), BridgeData and EpicKitchen-100 (Epic100). We follow Gu et al. (2023) to report FVD and KVD (↓ the lower the better). For each method, we also report the data type used for pre-training (if any) and whether it supports text conditioning. The best numbers are in bold black. Our method is shaded in `gray`. Rows 1–8 are cited from (Gu et al., 2023).

In inference, we use the DDIM (Song et al., 2020a) sampler to accelerate the process while leveraging the classifier-free guidance (CFG) per Equation 2. For the unconditional prediction, we follow the standard practice (Ho & Salimans, 2021) by replacing the text prompt with an empty string and passing it to the text encoder to obtain $\mathbf{y}$. However, we do not drop the initial frames as it empirically harms the generation quality.

## 4 Experiments

**Datasets.** We follow Gu et al. (2023) and test on three text-to-video datasets. **Something-Something-V2 (SSv2)** (Goyal et al., 2017) contains 168,913 training samples on daily human behaviors of interacting with common objects. **Bridge-Data-V1** (Ebert et al., 2021) includes 20,066 training samples in kitchen environments where robot arms manipulate objects. **Epic-Kitchen-100 (Epic100)** (Damen et al., 2020) consists of 67,217 training samples of human actions in first-person (egocentric) views. All datasets include short text descriptions of motions per video. In validation, we follow the implementation[1] by Gu et al. (2023) for a fair comparison. For SSv2, we select the first 2,048 samples in the validation set; for BridgeData, we split the dataset by 80% and 20% and use the latter for validation; finally, for Epic100, we directly adopt its validation split.

**Implementation Details.** We leverage the CogVideoX-2B (Yang et al., 2024b) T2V model in this work. We use its pre-trained weights[2] for initializing the 3D VAE, the T5 text encoder (Colin, 2020), and the DiT, while freezing these modules. We initialize the FCA attention layers from scratch for fine-tuning. Regarding the frame-wise text conditioning modules, we initialize its ViT (Dosovitskiy et al., 2020) vision encoder, as well as the CLIP (Radford et al., 2021) text tokenizer and projection layer from BLIP-2[3]. For each DiT layer, we initialize a frame-wise text conditioning module of one layer from scratch and train them alongside the FCA attention blocks.

For DiT, we use two initial frames on SSv2, and one initial frame on BridgeData and Epic100, as in (Gu et al., 2023). For the frame-wise text conditioning module, due to its architectural design that accepts only one image through cross-attention, we input the first frame regardless of the dataset. We train our model for 100,000 steps on SSv2, 25,000 steps on BridgeData, and 50,000 steps on Epic100. In inference, we set the CFG guidance scale $\lambda = 6.0$ and a sampling step of 50, following (Yang et al., 2024b). Further details are elaborated in Section B.1.

---

[1]https://github.com/seervideodiffusion/SeerVideoLDM
[2]https://huggingface.co/THUDM/CogVideoX-2b
[3]https://huggingface.co/Salesforce/blip2-itm-vit-g

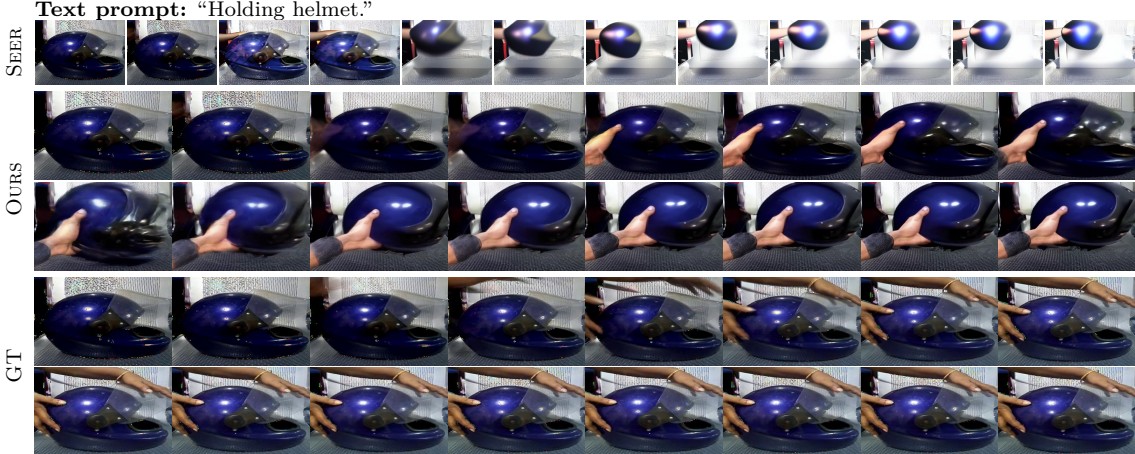

Figure 4: **Qualitative examples on video generation**. We compare our method with Seer (Gu et al., 2023), where we also show the ground truth (GT). For inference on Seer, we adhere to its configuration with a frame number of 12; while ours is of 16 (split into two rows). For illustration purposes, we replace the conditioning frames (first two) with the ground truth. See qualitative examples on other datasets in Section C.6. Note that the aspect ratios of the figures are slightly adjusted for demonstration purposes.

**Evaluation Settings.** Following (Gu et al., 2023), we assess our method on both machine and human evaluations. For machine evaluation, we include Fréchet Video Distance (FVD) and Kernel Video Distance (KVD) (Unterthiner et al., 2018). Both metrics are computed with a Kinetics-400 pre-trained I3D model (Carreira & Zisserman, 2017). See Section C.2 for discussions on human evaluation.

We additionally assess on VBench (Huang et al., 2024), which is a recent, well-regarded evaluation benchmark for video generation. However, we note that certain adaptations are required for us to benchmark on this metric. We refer readers to Section C.1 for details and our results.

All ablation studies are conducted on SSv2 with the FVD and KVD metrics. To more efficiently compare various setups, we perform all ablation studies for 20,000 steps.

## 4.1 Quantitative Results

**Comparison with Existing Methods.** As shown in Table 1, our method (row 9) surpasses previous methods in both the FVD and KVD metrics. Notably, compared to Seer (Gu et al., 2023) (row 8), we achieve an approximately 40% reduction in the FVD score on SSv2 and a 60% reduction on BridgeData. This suggests that our model generates predictions that are visually closer to the ground truth videos, which implies that FCA succeeds in harnessing the generative abilities of the pre-trained T2V model. On Epic100, we also acquire a better performance compared to the previous methods, albeit having an FVD score of above 150, which is noticeably higher than scores on the other two datasets. We note that compared to SSv2 and BridgeData where the camera angle is mostly fixed throughout a video, Epic100 is arguably the most challenging dataset among the three, as it contains egocentric views where the camera is attached to the performer (human being), thereby including more substantial view changes and movements of the environments (see Section C.6 for qualitative examples). This characteristic renders the video frames harder to predict.

**Comparison with Image-to-Video (I2V) Models.** Even though pre-trained video generative models are traditionally text-based; most recently, contemporary work has begun to support image conditions alongside the text. As such, we include discussions on the image-to-video (I2V) model in comparison to our method. We take CogVideoX1.5-5B-I2V[4] for the assessment. Compared to our base model (CogVideoX-2B), this model is of a larger parameter size and includes architectural improvements as noted by "1.5". It has

---

[4]https://huggingface.co/THUDM/CogVideoX1.5-5B-I2V

| Methods | Pre-trained with | FVD ↓ | KVD ↓ |
|---|---|---|---|
| **1** CogVideoX1.5-5B-I2V (Yang et al., 2024b) | image-video | 198.0 | 0.20 |
| **2** CogVideoX1.5-5B-I2V + LoRA | image-video | 352.6 | 0.45 |
| **3** FCA (Ours) | text-video | **67.7** | **-0.18** |

Table 2: **Comparison with pre-trained image-to-video (I2V) models.** We report CogVideoX1.5-5B-I2V (Yang et al., 2024b) with and without LoRA (Hu et al., 2022a) fine-tuning. Experiments on SSv2. The best numbers are in bold black. Our method is shaded in `gray`.

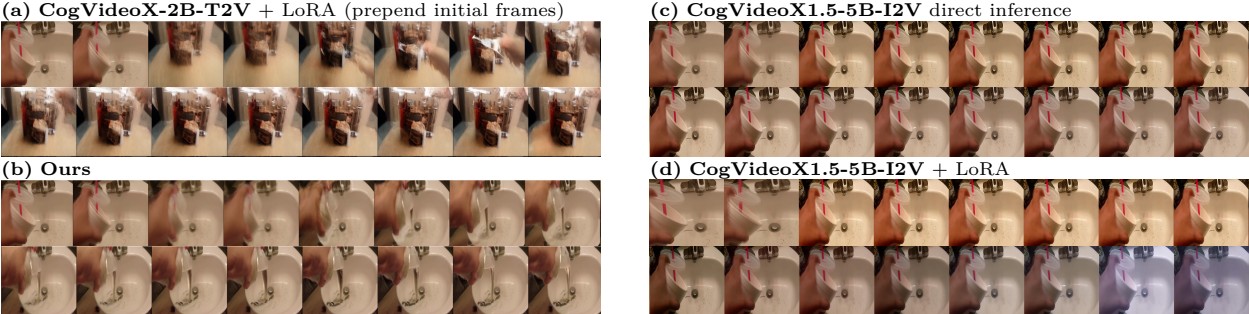

**(a) CogVideoX-2B-T2V** + LoRA (prepend initial frames)

**(b) Ours**

**(c) CogVideoX1.5-5B-I2V** direct inference

**(d) CogVideoX1.5-5B-I2V** + LoRA

Figure 5: **Comparison with LoRA (Hu et al., 2022a) fine-tuning and pre-trained I2V model (CogVideoX1.5-5B-I2V).** We replace the conditioning initial frames (first two) with the ground truth. Each sample is of 16 frames, split into two rows. Text prompt: *"Pouring something out of something"*.

been further trained from a T2V model to support a conditional frame in addition to the text (see Section 2). As shown in Table 2 (row 3 *vs.* 1), our method achieves a much better performance in FVD and KVD metrics, which is impressive since we use a model with a smaller, two billion parameter size, while competing with a more advanced architecture. We show that the I2V model often cannot predict the correct motions (Figure 5 c). Granted, this is not an entirely fair comparison because the I2V model is not trained on SSv2. However, we note that the performance does not improve, even degrades with LoRA fine-tuning (see discussions on LoRA in Section 4.3), as shown in Table 2 (row 2) and Figure 5 (d). This suggests the necessity of fine-tuning with a capable adaptation design for this task.

## 4.2 Qualitative Analysis

Figure 4 qualitatively compare our method with existing work. Compared to Seer (Gu et al., 2023), our method exhibits a stronger ability in temporal consistency and fidelity, demonstrating that we succeed in harnessing the power of the pre-trained video generative model. The examples also show that the model aligns well with the text. Interestingly, we illustrate a case where the generated contents differ from the ground truth, yet remain consistent with the input prompt. Such cases may not be accurately measured by the standard FVD and KVD metrics, which compute the similarity between the prediction and the ground truth. This partially demonstrates the necessity of extra quantitative evaluations, to this end, we refer readers to the analysis on human evaluation as well as VBench (Huang et al., 2024) (Section C).

## 4.3 Ablation Studies

In this section, we analyze the effects of our key design choices, as well as training tricks that are critical to the performance, as mentioned in Section 3.

**Fine-tuning Strategy** (`FT Strat`). We begin with validating our fine-tuning strategy, i.e., FCA, by comparing it with low-rank adaptation (LoRA) (Hu et al., 2022a) as the standard fine-tuning approach on generative models. Table 3 (rows 1, 2) shows our design achieves a much better performance in generation, where the FVD scores are reduced by over 100%. Note that both experiments are conducted without

| | FT Strat. | FTC | Attn. Mask | FVD ↓ | KVD ↓ |
|---|---|---|---|---|---|
| **1** | LoRA (Hu et al., 2022a) | ✗ | — | 354.9 | 0.78 |
| **2** | FCA | ✗ | — | 152.3 | 0.18 |
| **3** | FCA | uniform | ✗ | 157.2 | 0.34 |
| **4** | FCA | uniform+R | ✗ | 155.2 | 0.20 |
| **5** | FCA | layer-wise | ✗ | 117.6 | 0.14 |
| **6** | FCA | layer-wise | ✓ | **94.1** | **0.11** |

Table 3: **Ablation studies of our design choices.** Here, FT Strat. stands for fine-tuning strategy; FTC for frame-wise text conditioning design; Attn. Mask for attention mask. For FTC variants: "uniform" – one FTC module for all FCA attention layers; "+R" – further refine the embeddings through attention; "layer-wise" – one FTC module per FCA attention layer. We refer readers to Section B.4 for architectural illustrations of these variants. Experiments are conducted on SSv2 for 20,000 steps. The best numbers are in bold black. Our method is shaded in gray.

introducing other trainable modules into the pipeline for controlling variables. For LoRA fine-tuning, we append the latent of the initial frames to the token sequence.

The benchmark is corroborated with qualitative results in Figure 5 (a), where we show that LoRA fine-tuning often fails at fully capturing visual cues in the initial frames, which renders the predicted frames merely "look similar in colors" to the ground truth. Meanwhile, the generated content is often blurry, contains distortions, and does not adhere to the text conditions. Inferior results are also observed for the more advanced CogVideoX1.5-5B-I2V model with LoRA fine-tuning (d), where the model fails to learn meaningful motions, and merely attempts to repeat the conditioning frame. We note that in this particular case, the color shift is conjectured to be contributed by other training samples, as SSv2 contain assorted, and widely different color patterns/saturation/brightnesses across video samples. This suggests that LoRA's limited parameter space (Hu et al., 2022a) struggles at learning valid patterns from the sophisticated and large amounts of training data. As a result, it fails at adapting the pre-trained T2V model to this task. This observation in turn supports our choice of a more capable adaptation strategy that yields much better predictions (Figure 5 b).

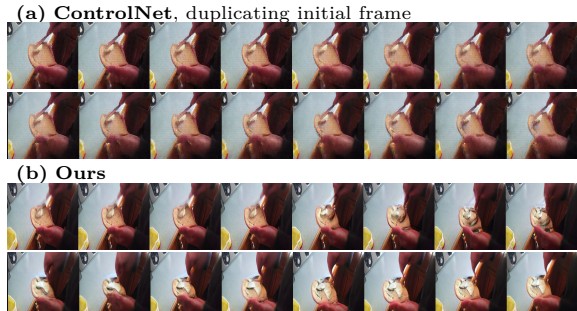

Figure 6: **Inserting initial frames.** We visualize ControlNet (Zhang et al., 2023a) *vs.* our approach (see Section 3.2). Experiments conducted without frame-wise text conditioning. Text prompt: *"Spreading margarine onto bread".*

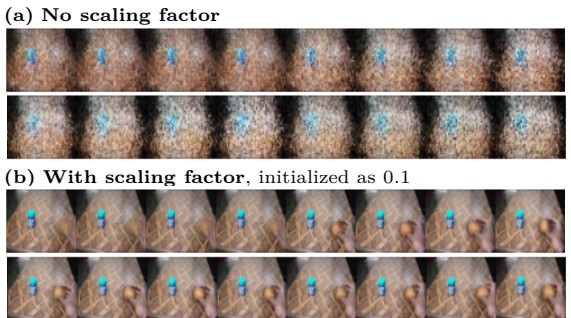

Figure 7: **FCA output integration.** We exemplify the significance of scaling factors. Text prompt: *"Holding potato next to Vicks VapoRub bottle".*

**Inserting Initial Frames.** Regarding our approach of inserting the initial frames into a T2V model (Section 3.2), we again validate it by comparison. Here, we showcase ControlNet (Zhang et al., 2023a) as a baseline — the widely adopted method of injecting spatial conditions into text-based generative models (Zhao et al., 2023; Hu & Xu, 2023). In Figure 6, we qualitatively demonstrate the results. Since ControlNet was originally developed on image generative models rather than video ones, we test two variants of the input inspired by recent work (Chen et al., 2024; Ye et al., 2023; Blattmann et al., 2023): either duplicating the initial frame along the frame dimension; or padding the initial frames with zero, i.e., black images. We

| | Fine-tuning with | FTC | Attn. Mask | Pre-Norm | FVD ↓ | KVD ↓ |
|---|---|---|---|---|---|---|
| **1** | FCA | layer-wise | ✓ | ✗ | 105.4 | 0.13 |
| **2** | FCA | layer-wise | ✓ | ✓ | **94.1** | **0.11** |

Table 4: **Ablation studies on pre-normalizing the frame-wise text conditioning $\mathbf{y}_{\text{frames}}$.** Experimented on SSv2 for 20,000 steps. The best numbers are in bold black. Our method is shaded in gray.

discover that in both cases, the model struggles to deviate from the conditioning visual cues in subsequent frames, which greatly hinders its ability to predict motions. As an example, we demonstrate the case of duplicating the initial frame in Figure 6 (a). Our intuition is that ControlNet strongly influences the latent embeddings via direct addition. This is acceptable for video-to-video editing or style transfer where the input condition is frame-wise matched to the intended output. However, it is not suitable for inserting the initial frames. We additionally note that one can observe mosaic-pattern artifacts — aside from poor motion predictions — by zooming into the ControlNet results, which is also undesired. These finding justifies the need for an approach more tailored to the task setup, and hence, our FCA module design (Figure 6 b).

**FCA Output Integration.** In Figure 7, we empirically illustrate the effect of scaling the FCA output before adding it back to the DiT branch, as discussed in Section 3.2. Note the training would fail without initializing the scaling factor to a relatively small value (i.e., 0.1). We conjecture that down-scaling effectively tunes down the influences of the added embeddings, thereby avoiding drastic changes that harm the training.

**Frame-wise Text Conditioning** (FTC). Table 3 rows 3–6 ablate our design choices of the frame-wise text conditioning module. Our key discovery is that separate, layer-wise conditioning modules perform better than using one such module to generate the frame-wise conditioning embeddings for the attention blocks of all FCA layers, even though the latter design is more commonly used with cross-attention (e.g., (Li et al., 2022; 2023a)). We note these two types of architectures as "layer-wise" and "uniform" in Table 3 and illustrate the architectural differences in Section B.5 (Figure 8) For the "uniform" architecture, we include two variants, with the difference being whether further to refine the frame-wise text embeddings throughout the FCA attention layers. As shown in Table 3 (rows 3–5), the layer-wise design outperforms others by a large margin. Notably, both "uniform" designs adversely harm the performance when compared to not adding the frame-wise text conditioning module (row 2), whereas "layer-wise" does not. The results fit our intuition in Section 3.3 that each DiT and FCA attention layer learns different semantics, and so should the conditioning embeddings.

**Frame-wise Attention Mask** (Attn. Mask). In Table 3 (rows 6 *vs.* 5) we show that the model learns better with the frame-wise attention mask, which enforces the frame-wise text conditioning modules in producing text conditions per frame, as introduced in Section 3.3. For a fair comparison, the two experiments adopt the same length in the text conditioning embeddings $\mathbf{y}_{\text{frames}}$.

**Pre-Normalizing Frame-wise Text Condition.** In Table 4, we show the need for pre-normalizing the frame-wise text embeddings before injecting it into the FCA cross-attention layer. Our intuition is that the operation mimics the pre-normalization, i.e., expert adaptive Layernorm (Peebles & Xie, 2023) performed in DiT layers, which helps bridge the potential modality gap between embeddings from different modules.

## 5  Limitations

When comparing with common adaptation strategies, we identify the main trade-offs of our method as the result of the increased parameter size introduced by the additional adaptation modules, which is more powerful in its capabilities, yet undoubtedly heavier than the conventional parameter-efficient adaptation methods, such as LoRA. This has brought two major implications. First, our method requires more computation resources, specifically, VRAM, which is a direct effect of the parameter count (Section B.2). Second, our method takes longer to train (Section B.1), as well as to inference — in practice, we observe the inference time increases by 21% when compared to a typical implementation of LoRA (Table 6). We provide the detailed hyperparameters related to the parameter count in Table 5.

In comparison with previous work, we note that our method adopts a more recent, however, relatively heavier text-to-video base model as opposed to, e.g., Seer (Gu et al., 2023), who adapted a pre-trained text-to-image model into video generation. While leveraging video generative models aligns with the community's progression, the text-to-video base model contains more parameters than the text-to-image model, for example, to learn additional temporal consistency across frames. Therefore, it also contributes to our method being heavier than the previous model.

## 6 Conclusion

We propose Frame-wise Conditioning Adaptation (FCA) for text-video prediction, which enables exploiting a pre-trained text-to-video generation model for text-video prediction. The FCA architecture is based on a parallel attention module applied to a diffusion transformer, alongside a frame-wise text conditioning module. This enables FCA to condition on both initial video frames and natural language. We have shown that the method greatly surpasses existing work, both in qualitative and quantitative results, and shared detailed practical findings and tricks to aid future research in this direction. Our ambition is that FCA might provide an indication of a potential avenue of exploration for scalable practical video generation methods.

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

# A    Appendix

# B    Additional Implementation Details

## B.1    Training and Inference

In Table 5, we report our detailed hyper-parameters for training. We note that our training configuration mostly inherits from the CogVideoX codebase[5].

We conduct our experiments on four NVIDIA A100 40G GPUs. In practice, we observe the VRAM usage at approximately 40GiB per GPU. The training takes approximately 24 hours per 10,000 steps with our batch size noted in Table 5) on SSv2. For inference, we leverage the `diffusers`'s pipeline[6] without altering its configurations. Specifically, we use the DDIM Song et al. (2020a) sampler with the guidance scale $\lambda$ set to 6.0, and the sampling step being 50. The evaluation on SSv2 (first 2,048 samples) takes around six hours on four NVIDIA A100 40G GPUs.

Our method is fine-tuned on a video resolution of $720 \times 480$ with the number of frames per video downsampled to 16 for all datasets. We follow the CogVideoX implementation[5] in resizing a given frame, while using the Seer Gu et al. (2023) codebase[7] for casting an arbitrary raw video to a fixed number of frames. Our choice of resolution is the lowest recommended by the pre-trained CogVideoX-2B model, we empirically confirm that setting the resolution to lower will irrecoverably disrupt the generation quality. Meanwhile, CogVideoX requires the frames to be divisible by 8. We note that this differs from the settings of previous work Gu et al. (2023), where the resolution is $256 \times 256$, with SSv2 being further downsampled to 12 frames. We note that our choice of a higher resolution and number of frames introduces a greater challenge to our method. However, it does not affect benchmarking on the FVD and KVD metrics, whose details are introduced in Section B.3.

In Table 6, we quantitatively compare FCA vs LoRA inference time obtained from the validation split. We observe that FCA increases the inference time per sample by 21%, which is to be expected given our increased parameter size (see Section B.2).

---

[5]https://github.com/THUDM/CogVideo
[6]https://huggingface.co/docs/diffusers/en/api/pipelines/cogvideox
[7]https://github.com/seervideodiffusion/SeerVideoLDM

| | Parameter | Value |
|---|---|---|
| **DiT** | Height | 480 |
| | Width | 720 |
| | Number of frames | 16 |
| | VAE temporal downsampling rate | 4 |
| | Effective number of frames | $16/4 = 4$ |
| | DiT text token length (T5 Colin (2020)) | 226 |
| **FTC** | Number of layers each | 1 |
| | Number of queries | $226 \times$ Eff. No. frames $= 904$ |
| **TRAINING** | Batch size per GPU | 1 |
| | Number of GPUs | 4 |
| | Gradient accumulation | 2 |
| | Effective batch size | $1 \times 4 \times 2 = 8$ |
| | Training steps (SSv2) | $100,000$ |
| | Training steps (BridgeData) | $25,000$ |
| | Training steps (Epic100) | $50,000$ |
| | Scheduler | `cosine_with_restart` |
| | Number of cycles | 1 |
| | Warmup steps | 200 |
| | Learning rate | 0.001 |
| | Optimizer | AdamW ($\beta_1 = 0.9, \beta_2 = 0.95$) |
| | Gradient clip | 1.0 |
| | Precision | `bfloat16` |

Table 5: **Hyper-parameters for fine-tuning.** FTC: frame-wise text conditioning module. Values computed from other parameters are shaded in `gray`.

| | Methods | Inference time for test split (2048 samples) (min) | Per-sample time (s) |
|---|---|---|---|
| **1** | LoRA | **1,172** | **34.34** |
| **2** | FCA (Ours) | 1,424 | 41.72 |

Table 6: **Inference efficiency.** Quantitatively, we observe that FCA increases the inference time per sample by 21%.

## B.2 Parameter Size Comparison

Table 7 demonstrates the parameter count for the state-of-the-art model (Seer (Gu et al., 2023)) preceding our method, as well as the parameter count for ours.

Meanwhile, we note that the remaining baseline methods shown in Table 1 are derived from Gu et al. (2023), where we cannot verify their training details or adaptations to this task. Nevertheless, we present the estimated total parameter counts for VideoFusion (Luo et al., 2023) and Tune-A-Video (Wu et al., 2023b) based on the publicly available knowledge, as they are two of the relatively recent methods that bear decent performance compared to Seer and ours.

Table 8 shows the trainable parameter count for LoRA and FCA (our module). We note that for LoRA on CogvideoX, we tested various rank values and our reported results (Section 2) are based on a rank of 256, which we discovered to be optimal (lower rank yields worse results, while higher ranks do not improve the performance).

## B.3 FVD and KVD Evaluation

We follow Gu et al. (2023) and adopt the evaluation code implemented by VideoGPT Yan et al. (2021). The benchmark requires pairs of prediction and ground truth videos with an equal number of frames, as well as the same resolution. Note that the prediction and ground truth are firstly resized into $224 \times 224$ regardless of the original resolution. This is because the evaluation is based on a pre-trained I3D model on Kinetics-400 Carreira & Zisserman (2017) that expects such inputs Unterthiner et al. (2018).

| Methods | Pre-trained with | Trainable Params | Base Model Params | Total Params |
|---|---|---|---|---|
| **6** VideoFusion (Luo et al., 2023)⋆ | text-video | – | – | 2.59B |
| **7** Tune-A-Video (Wu et al., 2023a)⋆ | text-image | – | – | 890M |
| **8** Seer (Gu et al., 2023)† | text-image | 405.90M | 890M | 405.90M + 890M = 1.3B |
| **9** Ours† | text-video | 1.0B | 2B | 1.0B + 2B = 3.0B |

Table 7: **Comparison of parameter counts across different video generation models.** ⋆ Values derived from publicly available knowledge. Specifically, for VideoFusion, pre-trained base model 2B + residual generator 0.59B = 2.59B; for Tune-A-Video, based on Stable Diffusion v1.5 as 890M. † Both methods apply additional trainable modules, i.e., additional parameters in addition to the base model parameters. Row numbers follow Table 1. The best-performing configuration (ours) is shaded in `gray`.

| Methods | Trainable Parameter Count |
|---|---|
| **1** LoRA (c.f. Table 2, row 2) | 0.27B |
| **2** FCA (Ours) | **1.04B** |

Table 8: **Comparison of trainable parameter counts.** FCA uses a larger but more expressive parameter subset than LoRA. The best-performing configuration (ours) is shaded in `gray`.

## B.4 Frame-wise Text Conditioning Design

In the main text Section 4.3 and Table 3, we have ablated various designs on the frame-wise text conditioning module (`FTC`). We introduce the implementation details as follows.

**Layer-wise Design.** We empirically determine this as the superior way of integrating the frame-wise text conditioning embeddings into the FCA attention layers, which is noted as "layer-wise" in Table 3, and demonstrated in Figure 2 along with the detailed architecture. In practice, each frame-wise text conditioning module is of one layer as reported in Table 5, where each layer contains a self-attention, a cross-attention, and a feed-forward block Li et al. (2022). We are unable to experiment with such modules of multiple layers under this setting due to the VRAM limitation.

**Uniform Design.** In Table 3 (rows 3–6), we include results on two variants of the frame-wise text conditioning design which yield inferior results, we label them as "uniform" (i.e., uniformly using one module for all layers) and "uniform+R" (further refining the module output through FCA attention layers), as shown in Figure 8. For these "uniform" designs which use only one such module in total, we are less constrained by the VRAM limitation compared to the above. To this end, we experiment with various numbers of layers. However, we empirically find this not critical for the performance. Indeed, as noted in the main text, Table 3, such "uniform" designs adversely harm the training, rendering it ineffectual in general.

## B.5 Implementation Details of Ablation Studies

**LoRA Fine-tuning.** In the main text Section 4.1 and 4.3, we have compared our method against LoRA fine-tuning. These LoRA experiments have been conducted on CogVideoX-2B (T2V) and CogVideoX1.5-5B-I2V. We adhere to the training configurations released in the official codebase[5] and make no changes to the parameters.

**ControlNet on DiT.** In the main text Section 4.3, we have conducted experiments on using ControlNet to inject the initial frames. We note that since ControlNet Hu & Xu (2023) was originally proposed on the U-Net Ronneberger et al. (2015) architecture, modifications are needed for it to be used on CogVideoX, which is based on DiT Peebles & Xie (2023). We experimented with two variants of the architecture applicable[8, 9], however, both yielded very similar inferior results, as demonstrated in the main text, Figure 6.

---

[8] https://github.com/TheDenk/cogvideox-controlnet
[9] https://github.com/Tencent/HunyuanDiT

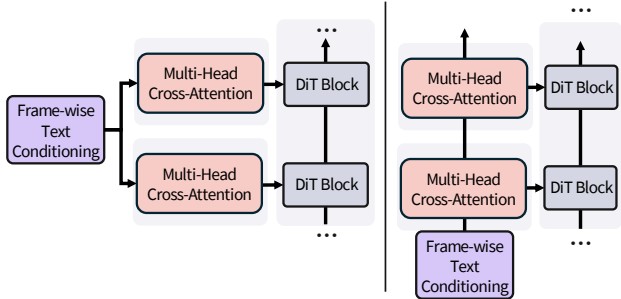

Figure 8: **Additional variants of designs to inject frame-wise text conditions into the FCA attention block.** This figure complements Figure 2 and results in the main text, Table 3 (rows 3 and 4). **Left:** Using one frame-wise text conditioning module for all FCA attention layers; we note this as "uniform" in the main text. **Right:** Further refining the embeddings through the FCA attention; we note this as "uniform+R" in the main text.

| | Methods | Background Consistency | Imaging Quality | Motion Smoothness | Overall Consistency | Subject Consistency | Temporal Flickering |
|---|---|---|---|---|---|---|---|
| **8** | Seer Gu et al. (2023) | 0.8655 | 0.3765 | 0.9172 | 0.1038 | 0.7301 | 0.8955 |
| **9** | FCA (Ours) | **0.8769** | **0.4279** | **0.9570** | **0.1965** | **0.8098** | **0.9461** |

Table 9: **VBench Huang et al. (2024) evaluation results.** We compare with Seer Gu et al. (2023). Results are reported on SSv2. Note that we use custom prompts (i.e., from the SSv2 validation set) for the evaluation, which differs from the standard VBench evaluation setup. Values are the higher the better. The best numbers are in bold black. Our method is shaded in gray.

## C  Additional Experimental Results

### C.1  VBench Evaluation

We additionally evaluate our model using VBench Huang et al. (2024), a recently introduced comprehensive benchmark designed for video generative models. Unlike standard video generation benchmarks that primarily evaluate unconditional video synthesis, VBench enables a more fine-grained analysis that directly aligns with our goal of producing temporally stable and semantically coherent video sequences.

**Evaluation Details.**  We specifically focus on six key dimensions: background consistency, imaging quality, motion smoothness, overall consistency, subject consistency, and temporal flickering. These dimensions are highly relevant to the text-video prediction task by collectively capturing both spatial quality and temporal stability.

We note that the standard VBench setup is designed for unconditional video generation, which assesses a model on a list of pre-defined prompts. While well-suited for benchmarking pre-trained general-purpose video generative models, it does not align directly with our fine-tuning task. As such, we opt for using custom prompts, i.e., the prompts from the validation set of SSv2 for the evaluation.

**Results.**  As shown in Table 9, our method outperforms Seer Gu et al. (2023) across all selected VBench dimensions. These results validate the effectiveness of our approach in improving both temporal coherence and per-frame quality in video prediction tasks. The substantial gains in consistency-related scores and motion smoothness indicate that our method successfully learns to maintain global coherence, while higher subject consistency confirms improved local stability. Meanwhile, we note that our method is better at adhering to the text prompt, as overall consistency is powered by ViCLIP (Wang et al., 2023b) to specifically assess text-visual alignment. These results highlight the robustness of our approach to produce high-quality, temporally consistent video predictions.

## C.2 Human Evaluation

**Evaluation Details.** Following previous work Gu et al. (2023), we conduct a human evaluation to assess the generation quality further. Specifically, we evaluate the SSv2 dataset, with 24 participants and 50 samples on three methods. We compare across our model, Seer Gu et al. (2023), and the more advanced CogVideoX1.5-5B-I2V Yang et al. (2024b). For each sample, we ask the participant to rate in integer, from 0 to 5 (inclusive, the higher the better) on three aspects: text alignment, temporal consistency, and visual fidelity. Text alignment focuses on the model's ability to generate motions that adhere to the input text. Temporal consistency assesses if the sample is a valid extension of the initial frames, and if the subsequent generated frames are consistent. Here, we replace the initial frames with the ground truth frames, thereby assessing if the immediate predicted frames seamlessly extend from these initial frames. Visual fidelity addresses the appearances of the predicted objects, as well as e.g., human hands throughout the video.

We supply the ground truth video to the participants as a reference to compare, and direct the participants to refrain from using the ratings 0 or 5 frequently, unless the sample is of extremely low or high quality. We allow the participants to review each sample multiple times and compare across each method.

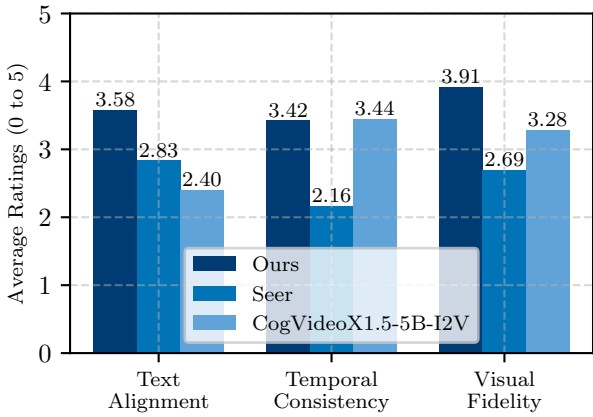

Figure 9: **Human evaluation results.** We report the average ratings (from 0 to 5 inclusive, the higher the better) among our method, Seer Gu et al. (2023), and CogVideoX1.5-5B-I2V Yang et al. (2024b).

**Results.** Figure 9 presents the average ratings for each method on the three aspects. Our method surpasses both Seer and CogVideoX1.5-5B-I2V on text alignment and visual fidelity, suggesting the effectiveness of our approach in fitting the pre-trained generative model to the task via fine-tuning. On text alignment, we note that CogVideoX1.5-5B-I2V receives the lowest rating, which is reasonable given that it has not been fine-tuned on the dataset. However, on visual fidelity, CogVideoX1.5-5B-I2V is more favored than Seer, which demonstrates the power of its pre-training. Our method receives a similar rating compared to CogVideoX1.5-5B-I2V on temporal consistency, while both surpass Seer. We note that the high rating of CogVideoX1.5-5B-I2V in this aspect is partly because the model often cannot render meaningful motions, and simply repeats the initial frames, thereby appearing consistent across time. This effect is discussed in the main text (Section 4.1) and demonstrated in Figure 5 (c). Overall, the human evaluation indicates the strength of our method across the three aspects.

## C.3 Additional Experiments on Wan2.1-1.3B

To demonstrate the generalizability of our adaptation strategy, we additionally test our method on Wan2.1-1.3B (Wan et al., 2025) and compare it with LoRA (Hu et al., 2022a) fine-tuning and the pre-trained model, as shown in Table 10.

| Methods | Additional Fine-tuning | FVD ↓ | KVD ↓ |
|---|---|---|---|
| **1** Wan2.1-1.3B I2V | N/A (pre-trained inference) | 155.12 | 0.10 |
| **2** Wan2.1-1.3B I2V | LoRA | 94.51 | -0.06 |
| **3** Wan2.1-1.3B T2V | FCA | **72.17** | **-0.11** |

Table 10: **Comparison of Wan2.1-1.3B (Wan et al., 2025) variants with different fine-tuning methods.** We report results for pre-trained inference and fine-tuned models using LoRA and FCA. The best scores are highlighted in bold, and our method is shaded in gray.

| Methods | Additional Fine-tuning | FVD ↓ | KVD ↓ |
|---|---|---|---|
| **1** CogVideoX1.5-5B I2V | DoRA (Liu et al., 2024) | 349.11 | 0.89 |
| **2** CogVideoX1.5-5B I2V | AdaLoRA (Zhang et al., 2023b) | 305.83 | 0.70 |

Table 11: **Comparison of CogVideoX1.5-5B T2V with different fine-tuning adapters.** We compare DoRA (Liu et al., 2024) and AdaLoRA (Zhang et al., 2023b) fine-tuning on the same base model. This table complements Table 2, row 2.

Note that we enlist the I2V[10] version for pure inference and LoRA fine-tuning, because the task of TVP requires one to input the initial frame as a condition alongside the text. Meanwhile, for our method, which is designed to adapt a T2V model, we apply it on the T2V pre-trained model[11].

Quantitatively, we note that our method is more effective than LoRA fine-tuning or plain inference of the pre-trained model, which corroborates our findings in our main paper and the performance of our method on DiT.

### C.4  Additional Experiments on variants of LoRA

We additionally test on variants of LoRA, namely, DoRA and AdaLoRA, which is demonstrated in Table 11. We note that these variants of LoRA perform similarly to the vanilla LoRA on the TVP task, in the sense that the generation quality is also less optimal.

### C.5  Quantitative Results on ControlNet Experiments

Table 12 demonstrates the quantitative results on the ControlNet-related experiments, which complements our discussion in Section 4.3.

### C.6  Additional Qualitative Examples

We include additional qualitative examples on SSv2 in Figure 10; on BridgeData in Figure 12 and Figure 13; and on EpicKitchen-100 (Epic100) in Figure 14 and Figure 15.

**Instruction Manipulation.**  In Figure 11, we showcase our method's ability to cope with instruction manipulation. Specifically, we perform inference on the fine-tuned model but manually alter the text prompt, e.g., "turning the camera left" to "turning the camera down"; or "putting egg in bowl" to "putting tomato in bowl". We show that our method can handle such variations and generate plausible predictions, indicating its superior generalizability.

### C.7  Limitations and Failure Cases

Figure 10 (h), Figure 13 (a), as well as Figure 15 illustrate several failure cases on different datasets.

---

[10]https://huggingface.co/alibaba-pai/Wan2.1-Fun-1.3B-Control
[11]https://huggingface.co/Wan-AI/Wan2.1-T2V-1.3B

| | Methods | Padding Strategy | FVD ↓ | KVD ↓ |
|---|---|---|---|---|
| **1** | ControlNet | Padding with zero (black) frames | 1073.43 | 2.18 |
| **2** | ControlNet | Duplicating the initial frame | 529.65 | 1.38 |

Table 12: **Results of different padding strategies on ControlNet performance.** Padding with duplicated initial frames significantly improves both FVD and KVD metrics. These results complement Section 4.3.

On SSv2, we note that the model may fail when facing object transformations or deformations, e.g., folding a mattress. Understandably, this poses a greater challenge compared to other common types of prompts, such as moving objects or the camera angle along a certain direction.

For BridgeData, we note that imperfection would sometimes arise when a certain object physically makes contact with another object (e.g., put something in a pot), thereby causing artifacts to be rendered in the scene. Such imperfections are often minor and occur at the ending frames, however, they are indeed failed attempts at maintaining the physical shape of an object.

On Epic100, we note that the prediction quality is usually worse than the other two datasets, which is corroborated by the quantitative metrics reported in the main text, Table 1. As shown in the failure cases, the generated contents are often blurry or contain artifacts. We note that this is partly because Epic100 is a dataset with egocentric views, which often includes frames with drastic camera motions (Figure 15). Combined with the fact that the kitchen environments are very diverse, this dataset is arguably the most challenging among the three.

# D  Potential Social Impacts

Our work is on video generation, which involves producing visual content based on user prompts. Our model is trained on samples involving the manipulation of objects by human beings or robot arms. This setup bears great potential in domains where such manipulations are of interest — such as action anticipations or planning in robotics and virtual/augmented reality. However, we note that our method could also be misused, which is a concern shared by most existing research on generative models. In our particular case, as a part of the training data (in SSv2 and EpicKitchen-100) portrays real-life human beings carrying out motions, our model might have learned to recreate specific human identities within such data. In addition, since our method is based on a large-scale, pre-trained text-to-video generative model, it shares many of the capabilities with the pre-trained model that could be used maliciously in creating and disseminating harmful or untrue content.

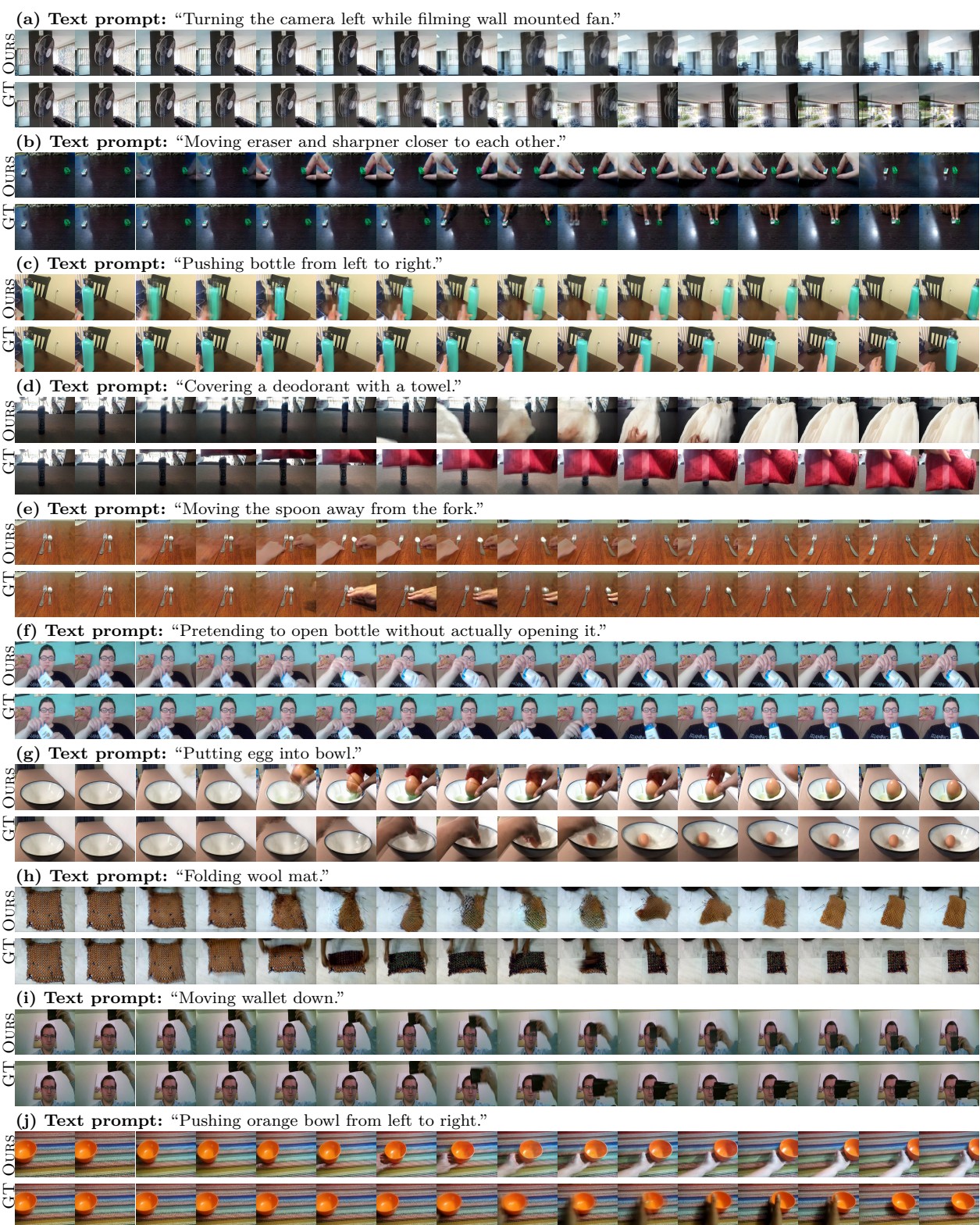

Figure 10: **Additional qualitative examples on Something-Something-V2 (SSv2).** Each sample is of 16 frames. We replace the initial frames (first two) with the ground truth.

**(a) Text prompt:** "Turning the camera left while filming wall mounted fan."

**Text prompt:** "Turning the camera right while filming wall mounted fan."

**Text prompt:** "Turning the camera up while filming wall mounted fan."

**Text prompt:** "Turning the camera down while filming wall mounted fan."

**(b) Text prompt:** "Putting egg into bowl."

**Text prompt:** "Putting tomato into bowl."

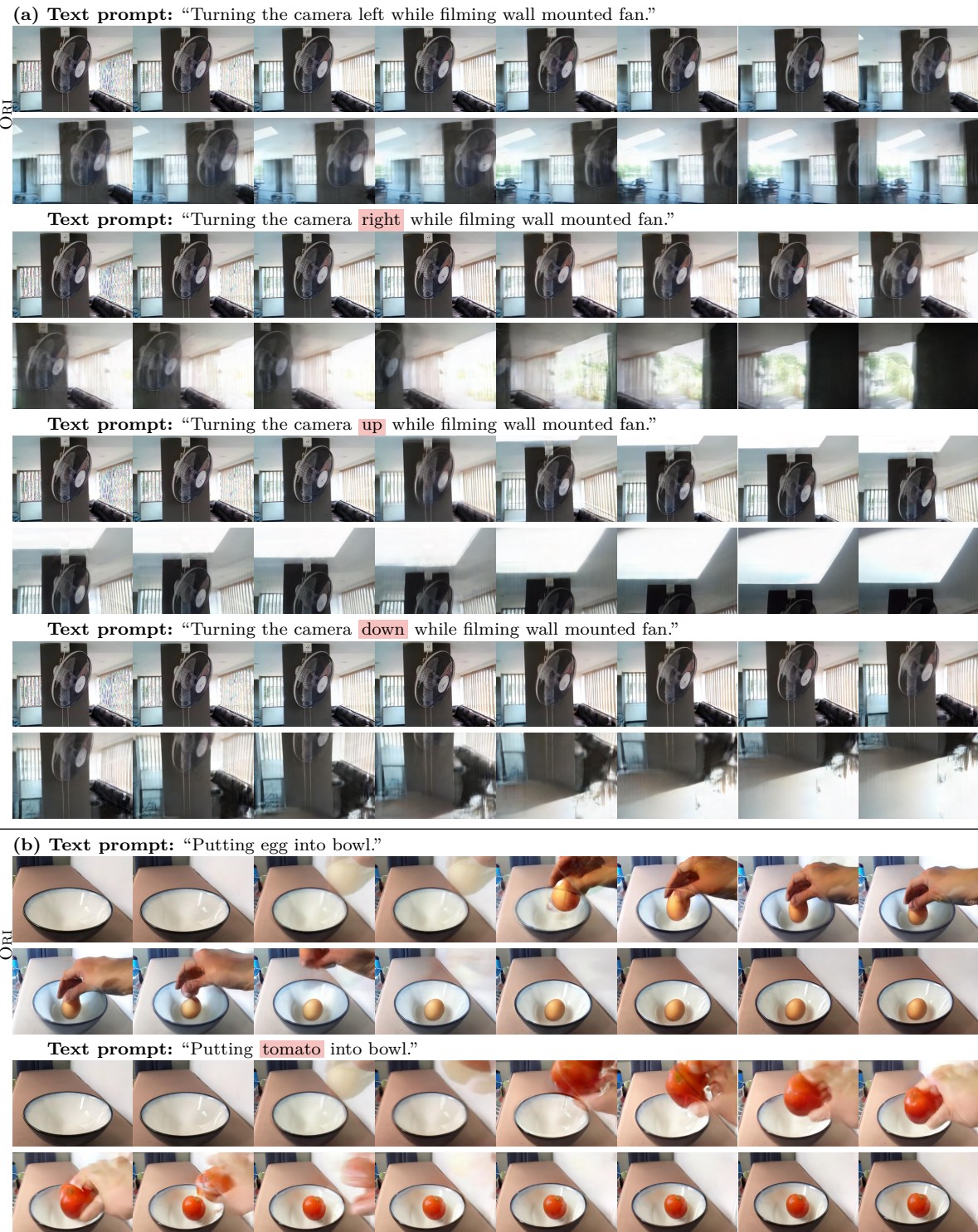

Figure 11: **Instruction manipulation on Something-Something-V2 (SSv2).** We manipulate the text prompt in inference to see if the model can predict diverse motions for a given initial frame. All frames are obtained from our model. ORI: the original text prompt. The modified phrases are shaded in red. Each sample is of 16 frames, split into two rows. We replace the initial frames (first two) with the ground truth.

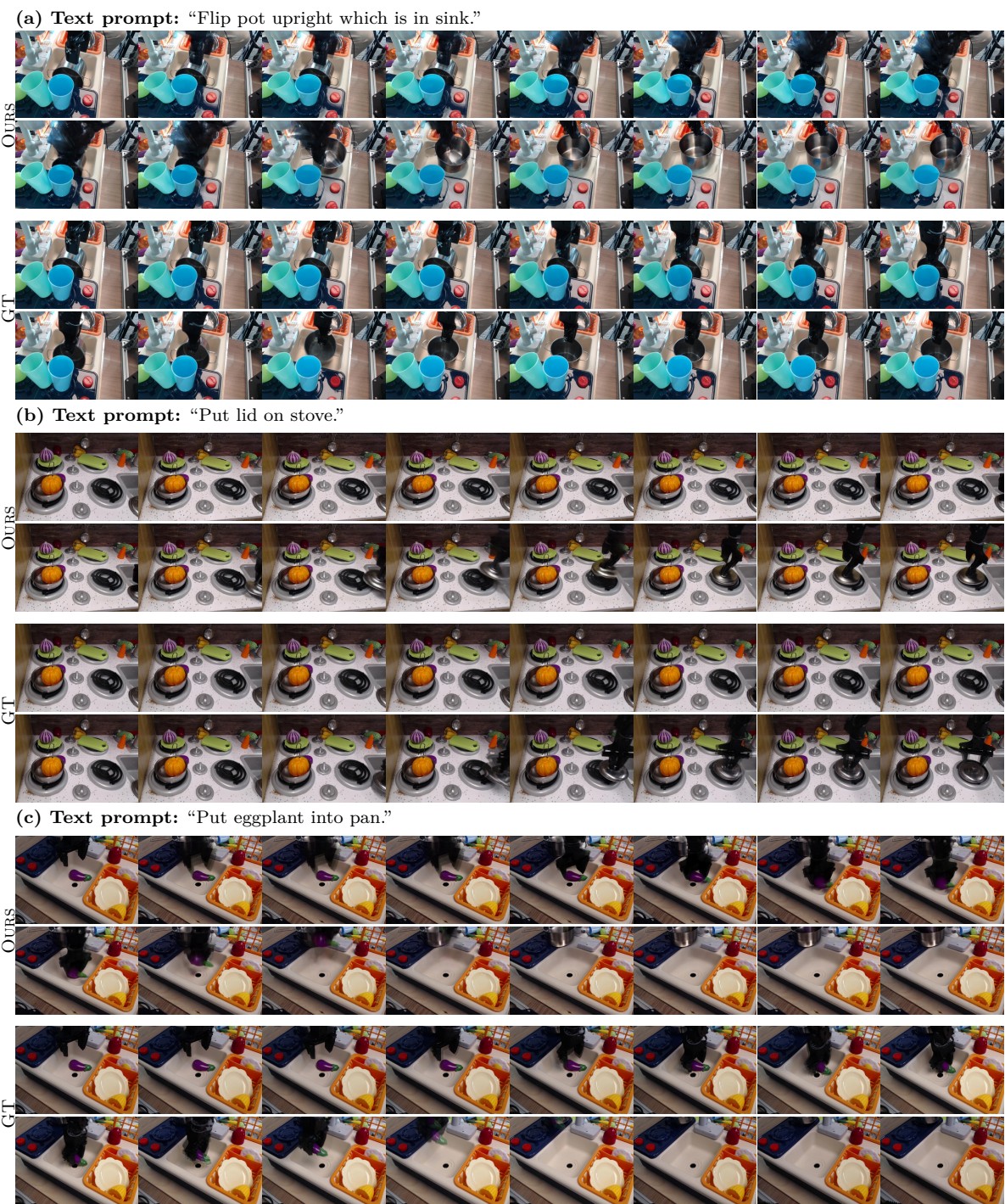

Figure 12: **Qualitative examples on BridgeData.** Each sample is of 16 frames, split into two rows for easier viewing of details.

**(d) Text prompt:** "Put detergent from sink into drying rack."

**(e) Text prompt:** "Put pepper in pot or pan."

**(f) Text prompt:** "Put sweet potato in pot."

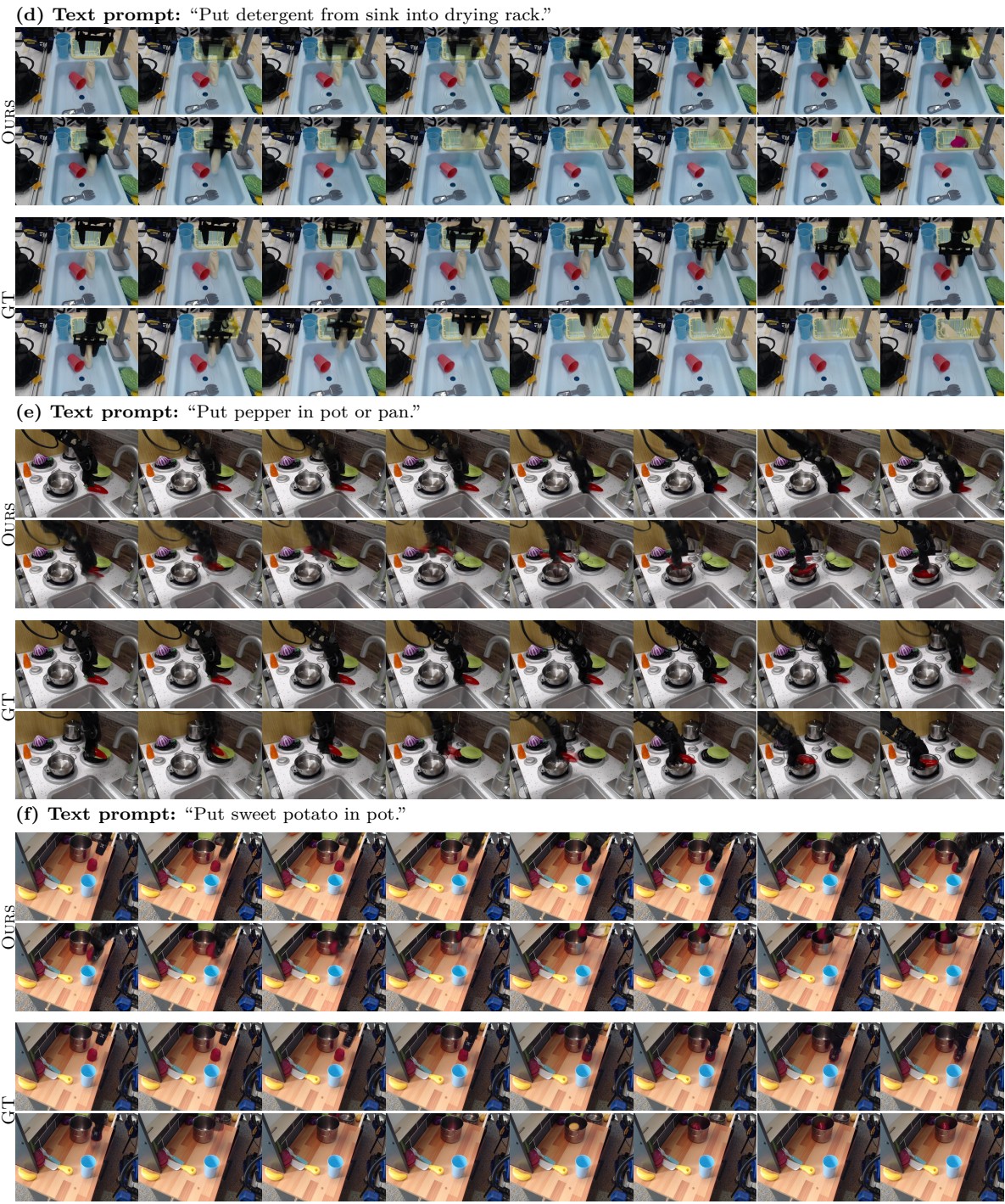

Figure 13: **Qualitative examples on BridgeData (cont.).** Each sample is of 16 frames, split into two rows for easier viewing of details.

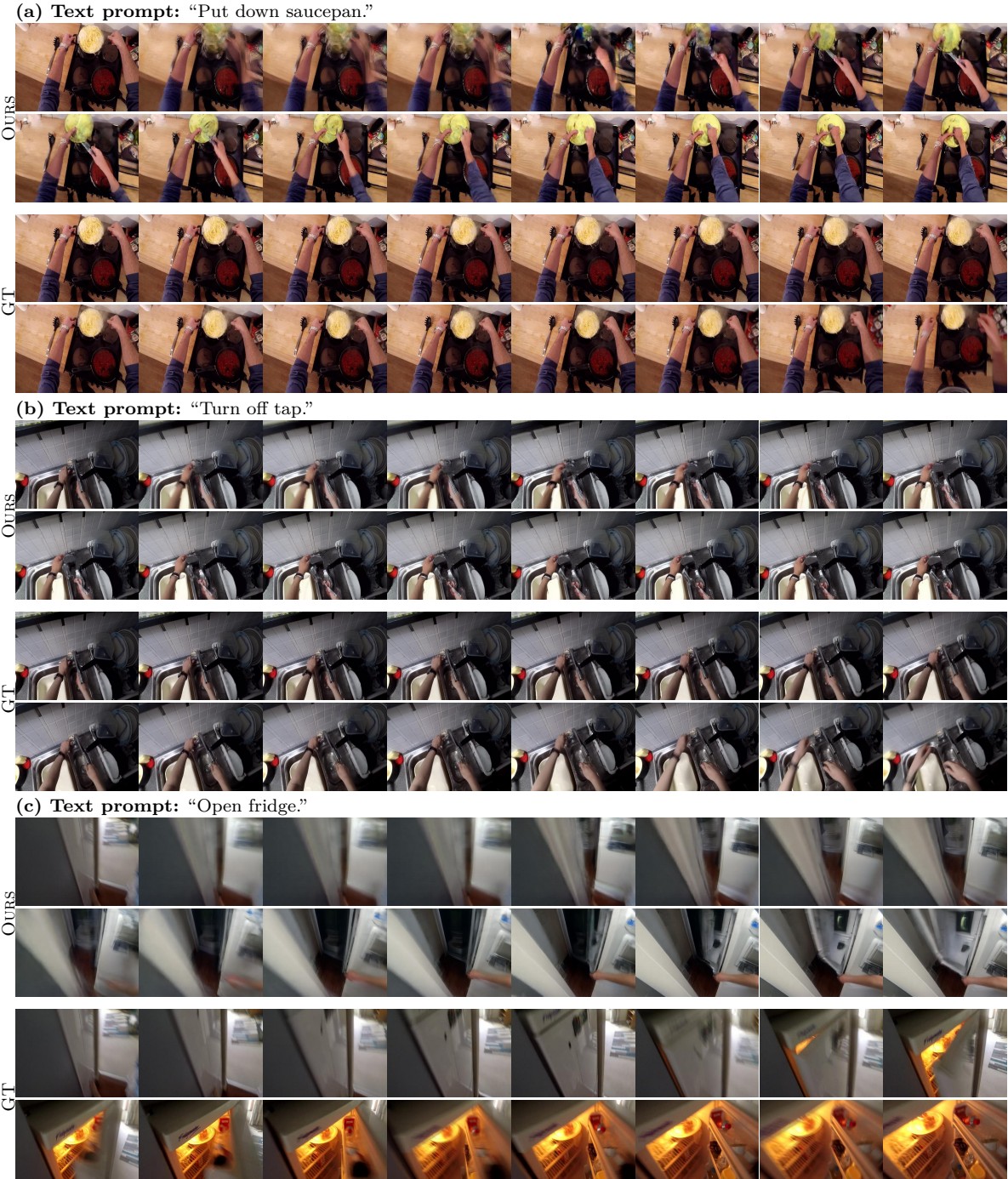

Figure 14: **Qualitative examples on EpicKitchen-100 (Epic100).** Each sample is of 16 frames, split into two rows.

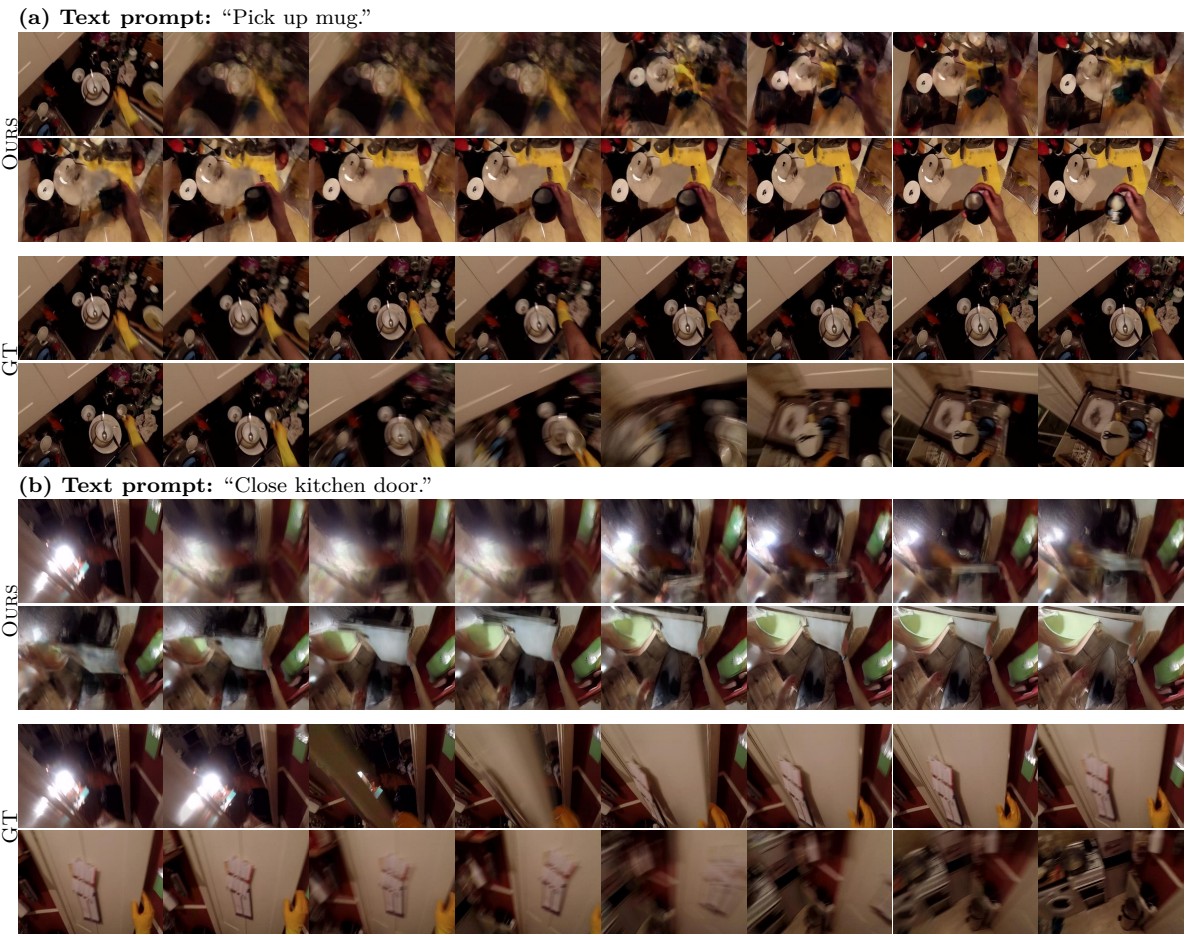

Figure 15: **Qualitative examples on EpicKitchen-100 (Epic100) (cont.).** We show several failure cases. Note the drastic change of view due to camera movements in the ground truths (appearing at the end of the video clips), making learning motions challenging. Each sample is of 16 frames, split into two rows.

