# OpenReview forum: "Frame-wise Conditioning Adaptation for Fine-Tuning Diffusion Models in Text-to-Video Prediction"
_TMLR — Accepted by TMLR_

### Review · Reviewer_sKjJ · 2025-09-13

**Summary Of Contributions:**

finetuning pre-trained text-to-video models using LoRA yields undesirable results. To address this, the authors propose an adaptation-based strategy, Framewise Conditioning Adaptation (FCA) to inject more context into the models embedding space. This strategy generates frame-wise text embeddings from input text as well as utilize the initial video frames, acting as an additional condition for generation.


Strengths:
- Method is well motivated and presented in a simple manner
- Empirical results are strong (large FVD/KVD improvements vs Seer et al.) and ablations credibly support the design choices
- Ablation studies provide a better understanding about FCA's contribution

Weaknesses:
- Generality beyond the object manipulation task is unclear from the results. performance on broader motion regimes is unclear.
- While baseline coverage is reasonable, comparisons with more recent T2V/DiT variants from 2024-2025 would strengthen claims, given the fast-moving landscape.

**Audience:**

Yes

**Audience Explanation:**

The proposed method, in my view, showcases how effective engineering can extend the usefulness of existing Text-Video Prediction models.

**Broader Impact Concerns:**

Yes, the authors mention the broader societal impacts in Appendix D.

**Claims And Evidence:**

Yes

**Claims Explanation:**

Yes, the claims made in the paper are supported. The authors provide qualitative and quantitative results showing LoRA inadequacy for Text-Video Prediction, technical design choices, human evaluations, along with failure case analysis.

**Requested Changes:**

- Please add Open-Sora and VideoCrafter2 as same-backbone baselines, each with LoRA-tuned vs FCA-tuned variants
- It would be great to also compare against other adaptation-based methods (AdaLoRA, DoRA)

---

> ### Author Response · Authors · 2025-10-07
> **Response to reviewer sKjJ**
>
> > Generality beyond the object manipulation task is unclear from the results. performance on broader motion regimes is unclear.
>
> We wish to note that we closely adhere to previous and pioneer work in the experiment setups for the task of text-video prediction (TVP), along with the datasets used.
>
> We agree with the reviewer that the current task setup focuses on manipulation prediction, which we have discussed at the beginning of our paper (Section 1).
>
> While our model design could potentially be viewed as task-agnostic, and could potentially serve as an inspiration for other tasks and domains, we have not claimed such generalizability due to the reason that this work predominantly focuses on the task of TVP.
>
> > While baseline coverage is reasonable, comparisons with more recent T2V/DiT variants from 2024-2025 would strengthen claims, given the fast-moving landscape.
> > Please add Open-Sora and VideoCrafter2 as same-backbone baselines, each with LoRA-tuned vs FCA-tuned variants
>
> We have tested our method on Wan2.1 and compared it with LoRA fine-tuning and the pre-trained model, which addresses the reviewer's concern about having more recent base models from the 2024-2025 era.
>
> Our choice of using the Wan2.1 model is based on the fact that, at the time of writing, it is one of the most powerful, recent video generation models that are open-source.
>
> We notice that the reviewers made two suggestions for testing. Between them, VideoCrafter2 is based on SD U-Net rather than Diffusion Transformer (DiT), where our method is designed for DiT. Meanwhile, Open-Sora is less recent and less performant compared to Wan2.1.
>
> We also wish to note that, given the time constraint of the rebuttal, we were unable to test on multiple base models.
>
> The results are as follows.
>
> |  | Method | Additional Fine-tuning | FVD | KVD |
> | :------- | :------- | :------- | :------: | -------: |
> | 1 | Wan2.1-1.3B I2V    | N/A (pre-trained inference)  | 155.12 | 0.10 |
> | 2 | Wan2.1-1.3B I2V  | LoRA | 94.51 | -0.06 |
> | 3 | Wan2.1-1.3B T2V  | FCA | 72.17 | -0.11 |
>
> Note that we enlist the I2V[a] version for pure inference and LoRA fine-tuning, because the task of TVP requires one to input the initial frame as a condition alongside the text. Meanwhile, for our method which is designed to adapt a T2V model, we apply it on the T2V pre-trained model[b].
>
> Quantitatively, we note that our method is more effective than LoRA fine-tuning or plain inference of the pre-trained model, which corroborates our findings in our main paper and the performance of our method on DiT.
>
> The above results are updated in Section C.3.
>
> ---
> *[a] https://huggingface.co/alibaba-pai/Wan2.1-Fun-1.3B-Control/tree/main*
>
> *[b] https://huggingface.co/Wan-AI/Wan2.1-T2V-1.3B*
>
> > It would be great to also compare against other adaptation-based methods (AdaLoRA, DoRA)
>
> We have conducted said experiments on AdaLoRA and DoRA, please see below.
>
> |  | Method | Additional Fine-tuning | FVD | KVD |
> | :------- | :------- | :------- | :------: | -------: |
> | 1 | CogVideoX1.5-5B I2V   | DoRA  | 349.11 | 0.89 |
> | 2 | CogVideoX1.5-5B I2V | AdaLoRA | 305.83 | 0.70 |
>
> We note that these variants of LoRA perform similarly to the vanilla LoRA on the TVP task, in the sense that the generation quality is also less optimal.
>
> The above results are updated in Section C.4.

---

### Review · Reviewer_bhJ4 · 2025-09-14

**Summary Of Contributions:**

The authors propose a fine-tuning method that extends text-to-video models to perform text-video prediction. In particular, they introduced Frame-wise Conditioning Adaptation (FCA), which can be inserted into a DiT block to allow initial frames as extra condition.

**Additional Comments:**

- “This is probably because the limited amount of tunable parameters of LoRA is insufficient to cope with the quantity and complexity of the training data”. This claim is weakly supported. In my opinion, FCA is more complex than LoRA and so the fact that it performs better than LoRA might tell us that FCA is more suitable for the task, but does not tell us about why LoRA fails.

**Audience:**

Yes

**Audience Explanation:**

I believe the text-video prediction problem is interesting and the paper establishes useful insights and baselines for the task.

**Claims And Evidence:**

Yes

**Claims Explanation:**

**Claim 1:**
- Existing TVP methods adapted from text-to-image (T2I) models lack continuity and quality.
- Evidence: Table 1 quantitatively shows that methods like Seer or Tune-A-Video can yield higher FVD/KVD (temporal consistency metrics) compared to FCA.


**Claim 2:**
- Standard fine-tuning with LoRA fails for TVP on T2V models.
- Evidence: Ablation experiments (Table 3, Figure 5) demonstrate LoRA produces blurry, distorted outputs that poorly adhere to text, with FVD scores significantly worse than FCA

**Claim 3:**
- The proposed Frame-wise Conditioning Adaptation (FCA) is an effective fine-tuning strategy for T2V models.
- Evidence: FCA integrates frame-wise conditioning modules and parallel attention while freezing the main DiT. Ablation studies show FCA outperforms LoRA and ControlNet baselines (Figures 5–6).

**Claim 4:**
- Incorporating initial frames as conditions improves temporal consistency in TVP.
- Evidence: Using frozen DiT embeddings of initial frames (Figure 2) leads to smoother video predictions. Comparisons against ControlNet baselines (Figure 6) show FCA handles motion prediction better, while ControlNet struggles with overfitting to static inputs

**Requested Changes:**

- Can the authors add parameter count for each model in Table 1?
- Can the authors add trainable parameter count for FCA and LoRA?
- For TVP task, I think that both video generation quality and text alignment should be equally important. However, the majority of the paper just discusses the first metric. On the other hand, the evaluation seems to be weak. I think an evaluation of more than 15 samples is needed to make it more persuasive. Also, I even think this section should be moved to the main text, not the Appendix.
- I like the ControlNet comparison, but I think it can be stronger with quantitative results.

---

> ### Author Response · Authors · 2025-10-07
> **Response to reviewer bhJ4 (part-1)**
>
> > Can the authors add parameter count for each model in Table 1?
>
> Please see below on the parameter count for the state-of-the-art model (Seer [Gu et al., 2023]) preceding our method, as well as the parameter count for ours.
>
> Meanwhile, we note that the remaining baseline methods shown in Table 1 are derived from Gu et al. (2023), where we cannot verify their training details or adaptations to this task. Nevertheless, we present the estimated total parameter counts for VideoFusion (Luo et al., 2023) and Tune-A-Video (Wu et al., 2023a) based on the publicly available knowledge, as they are two of the relatively recent methods that bear decent performance compared to Seer and ours.
>
> |  | Method | Pre-trained with | Trainable Parameter Count | Base Model Parameter Count | Total Parameter Count |
> | :------- | :------- | :------- | :------: | :-------: | -------: |
> | 6 | VideoFusion (Luo et al., 2023)*   | text-video  | --   | -- | pre-trained base model 2B + residual generator 0.59B = 2.59B |
> | 7 | Tune-A-Video (Wu et al., 2023a)*  | text-image | --   | --  | Stable Diffusion v1.5: 890M |
> | 8 | Seer (Gu et al., 2023)^ | text-image   | 405.90M  | Stable Diffusion v1.5: 890M  | 405.90M + 890M = 1.3B |
> | 9 | Ours^   | text-video  | 1.0B  | CogVideoX-2B: 2B  | 1.04B + 2B = 3.0B |
>
> *Values derived from the papers and repositories.
>
> ^Both methods apply additional trainable modules, i.e., additional parameters in addition to the base model parameters.
>
> *Row numbers follow Table 1.*
>
> This has been updated in Section B.2.
>
> > Can the authors add trainable parameter count for FCA and LoRA?
>
> Below, we show the trainable parameter count for LoRA and FCA (our module):
>
> |  | Method | Trainable parameter count |
> | :------- | :------- | :-------: |
> | 1 | LoRA (c.f. Table 2 row 2) | 0.27B |
> | 2 | FCA (ours)  | 1.04B |
>
> We note that for LoRA on CogvideoX, we tested various rank values and our reported results (Table 2) are based on a rank of 256, which we discovered to be optimal (lower rank yields worse results, while higher ranks do not improve the performance).
>
> This has been updated in Section B.2.
>
> > For the TVP task, I think that both video generation quality and text alignment should be equally important. However, the majority of the paper just discusses the first metric.
>
> We note that:
>
> - The FVD/KVD metric assesses the adherence of the generated video when compared to the ground-truth video. As the ground-truth video illustrates the actions denoted by the text, this metric inherently also assesses text alignment.
> - We additionally test on VBench (Huang et al., 2024) (Appendix C.1, Table 9), which contains, among multiple categories, "overall consistency"* -- which specifically evaluates the text alignment quality powered by ViCLIP (Wang et al., 2023b).
>
> ---
> *The naming of this category is slightly confusing as it does not explicitly mention "text alignment"; however, it is indeed addressing this aspect. We refer readers to the VBench (Huang et al., 2024) paper for details.*
>
> > On the other hand, the evaluation seems to be weak. I think an evaluation of more than 15 samples is needed to make it more persuasive. Also, I even think this section should be moved to the main text, not the Appendix.
>
> We believe the reviewer is referring to the human evaluation in this comment. To this end, we perform additional human evaluations on 35 samples, which, combined with the original 15 samples, total 50 samples.
>
> To maintain consistency with the existing scores, we purposefully make sure to seek the previous participants of the evaluation. However, despite our best efforts, three participants are no longer available. We therefore introduce new participants in their place.
>
> The updated human evaluation does not change our conclusion. We update the new average scores in Figure 9.
>
> We will consider moving this entire section to the main text if the space allows.
>
> > I like the ControlNet comparison, but I think it can be stronger with quantitative results.
>
> Please see the values below for the ControlNet-related experiments we tested.
>
> |  | Method | Padding Strategy | FVD | KVD |
> | :------- | :------- | :------- | :------: | -------: |
> | 1 | ControlNet   | Padding with zero (black) frames  | 1073.43   | 2.18 |
> | 2 | ControlNet  | Duplicating the initial frame | 529.65  | 1.38  |
>
> As illustrated above, we notice that using ControlNet as a means of injecting the initial frame severely damages the model's generation ability, which is discussed in the main paper Section 4.3, where we note that the model will struggle with deviating from the per-frame condition that is injected via ControlNet.
> This results in the generated output being completely non-usable, with very high FVD and KVD scores (which are the lower the better).
>
> See updated Section C.5.
>
> *cont. below*

---

> ### Author Response · Authors · 2025-10-07
> **Response to reviewer bhJ4 (part-2)**
>
> > “This is probably because the limited amount of tunable parameters of LoRA is insufficient to cope with the quantity and complexity of the training data”. This claim is weakly supported. In my opinion, FCA is more complex than LoRA, and so the fact that it performs better than LoRA might tell us that FCA is more suitable for the task, but does not tell us about why LoRA fails.
>
> We have revised this statement in Section 1.

---

### Review · Reviewer_Yjeu · 2025-09-22

**Summary Of Contributions:**

The authors introduce a novel fine-tuning method (and architecture) for improving sample quality of text-to-video (T2V) generation models, specifically focusing on diffusion transformer-based models. They're adaptation fine-tuning (adaptation) approach, named Frame-wise Conditioning Adaptation (FCA) uses an auxiliary attention-based architecture to provide frame-wise text embeddings for each frame, which improves upon existing techniques for fine-tuning T2V generation models. The authors also provide a thorough empirical evaluation of their newly proposed approach to validate its usefulness for fine-tuning T2V generation models. Through this empirical evaluation, the authors show that FCA (their approach) is a dominant fine-tuning approach for improving video generation quality, in comparison to the baselines considered in this work.

**Strengths:**

- The authors propose a novel fine-tuning method, FCA, for T2V diffusion transformer models. The method is elegant and can incorporate conditions from both video frames and natural language within its fine-tuning procedure.
- The authors provide a comprehensive empirical evaluation and demonstrate that their proposed fine-tuning method, FCA, improves sample quality of T2V generation, and outperforms all counterpart baseline approaches (which were considered in this work).
- The manuscript is generally well written and easy to follow, making it easy to understand the central contributions of this work.

**Weaknesses:**

- My understanding is that the authors consider only 1 pre-trained text-to-video model in their experiments (CogVideoX-2B). Since this work proposes (and claims) a general method for fine-tuning pre-trained T2V diffusion models, it would be useful to see experiments/results for 1-3 other models. Possibly, it is sufficient to pick a single experiment and show performance for different models.
- No explicit mention of the limitations in the main text. For example, it sounds like FCA requires the fine-tuning of an additional attention-based architecture. How does this compare to existing approaches? It is valuable to mention some of the trade-offs (if any) and limitations regarding the increased complexity of fine-tuning the proposed method in the main text/conclusion.
- Only a minor weakness, some portions of the manuscript could use further clarification and improved organization. See questions and minor comments below.

**Additional Comments:**

**Questions:**

- In several of the figures showing qualitative results, the authors compare to "ground truth" frames. How are the ground truth frames acquired in this setup? Is this based on the training data, i.e. for each text prompt, e.g. like in Figure 4, "Holding helmet", is there a corresponding set of frames (a video) that the model uses during training?
- How does FCA affect inference-time (if at all)?

Minor comments:

- Figure 1 is never referenced in the text.
- It would be helpful for the reader to move figures closer to where they are referenced in the text (e.g. Figure 2 and Figure 3).

**Audience:**

Yes

**Audience Explanation:**

Yes, I believe the findings of this work would be of interest to TMLR's vision and video generation audience. This paper tackles the problem of improving fine-tuning of pre-trained diffusion transformers for T2V generation. I believe this is an interesting a relevant problem in the domain.

**Broader Impact Concerns:**

I have no broader impact concerns.

**Claims And Evidence:**

Yes

**Claims Explanation:**

In general, yes. The authors conduct a thorough empirical evaluation of their proposed approach by considering various experimental settings, several baselines, and show both quantitative and qualitative results. Because of this, I believe the claims made in this submission are sufficiently supported.

**Requested Changes:**

Apart from minor clarifications (see comments below), my requested changes are not pertinent. However, I believe they could help improve the overall quality of this work. Specifically, applying FCA to an additional pre-trained T2V model and adding a more explicit discussion of limitations and future work in the main text (conclusion).

---

> ### Author Response · Authors · 2025-10-07
> **Response to reviewer Yjeu**
>
> > My understanding is that the authors consider only 1 pre-trained text-to-video model in their experiments (CogVideoX-2B). ... Possibly, it is sufficient to pick a single experiment and show performance for different models.
>
> We have further tested our method on Wan2.1 and compared it with LoRA fine-tuning and the pre-trained model.
>
> Our choice of using the Wan2.1 model is based on the fact that, at the time of writing, it is one of the most powerful, recent video generation models that are open-source.
>
> We also wish to note that, given the time constraint of the rebuttal, we were unable to test on multiple base models.
>
> The results are as follows, which hopefully strengthen the utility of our method.
>
> |  | Method | Additional Fine-tuning | FVD | KVD |
> | :------- | :------- | :------- | :------: | -------: |
> | 1 | Wan2.1-1.3B I2V    | N/A (pre-trained inference)  | 155.12 | 0.10 |
> | 2 | Wan2.1-1.3B I2V  | LoRA | 94.51 | -0.06 |
> | 3 | Wan2.1-1.3B T2V  | FCA | 72.17 | -0.11 |
>
> The above results are updated in Section C.3.
>
> Meanwhile, we wish to note that while our adaptation method could potentially be viewed as task-agnostic, i.e., generalizable towards other domains and tasks, and we indeed hope that our method serve as an inspiration for the community that potentially goes beyond the task of text-video prediction (TVP), we have not claimed such a general property as the highlight of this paper as our main focus is still on the TVP task.
>
>
> > No explicit mention of the limitations in the main text. For example, it sounds like FCA requires the fine-tuning of an additional attention-based architecture. How does this compare to existing approaches? It is valuable to mention some of the trade-offs (if any) and limitations regarding the increased complexity of fine-tuning the proposed method in the main text/conclusion.
>
> Please see the revised manuscript, Section 5.
>
> > In several of the figures showing qualitative results, the authors compare to "ground truth" frames. How are the ground truth frames acquired in this setup? Is this based on the training data, i.e. for each text prompt, e.g. like in Figure 4, "Holding helmet", is there a corresponding set of frames (a video) that the model uses during training?
>
> The train/test-split is produced by the authors of the datasets (Goyal et al., 2017; Ebert et al., 2021; Damen et al., 2020), and we, as well as previous work, follow them. By definition, a particular sample within the test-split must not exist in the train-split, otherwise it constitutes a data leakage.
>
> For e.g., a test sample with the text of "holding helmet", we would expect the training set to contain similar prompts (and accompanying videos), such as "holding pen" or "holding potato". It is the objective of this task to train the model, through learning on these samples, such that it learns the concept of motions and how to predict them.
>
> > How does FCA affect inference-time (if at all)?
>
> Below, below we compare FCA vs LoRA inference time:
>
> |  | Method | Inference time for the test split (2048 samples) (minutes) | Inference time per sample (second) |
> | :------- | :------- | :------- |  :------- |
> | 1 | LoRA   | 1,172  | 34.34 |
> | 2 | FCA (ours)  | 1,424 | 41.72 |
>
> Quantitatively, we observe that FCA increases the inference time per sample by 21%.
>
> This has been updated in Section B.1.
>
> > Figure 1 is never referenced in the text.
> > It would be helpful for the reader to move figures closer to where they are referenced in the text (e.g. Figure 2 and Figure 3).
>
> We have revised these in Section 1, and rearranged the figures where it is possible to do so (some are constraints by typesetting of latex).

---

### Author Response · Authors · 2025-10-07
**Response to reviewers' comment**

We thank the reviewer for their feedback. We have thus updated our manuscript, revised parts are marked in blue.

For detailed responses, please see below in our replies. In each reply, we have referred to changes made in the manuscript that correspond to it.

---

### Decision · Action_Editor_KJYS · 2025-11-15

**Recommendation:** Accept as is

**Audience:**

Yes

**Audience Explanation:**

The paper addresses a well-defined gap in the current landscape of text-guided video generation models, and the findings are likely of interest to computer vision and multimodal AI researchers.

**Claims And Evidence:**

Yes

**Claims Explanation:**

The paper introduces FCA (framewise conditioning adaptation), an approach to improve the performance of text-video prediction (TVP) models. At a high level, the motivation of this approach starts with the observation that existing pre-trained models for TVP exhibit continuity/quality artifacts, and standard LoRA-based adaptation approaches do not give very good results. To remedy this, the authors propose frame-wise text conditioning at each layer of a diffusion transformer, which is inspired by the Querying transformer trick used in BLIP-2. The authors show improvement in benchmark performance on 3 text-to-video datasets, as well as on a more recent benchmark (VBench) that enables fine-grained analysis.

Overall the paper reads very nicely and the findings should be interested to the broader computer vision community. The reviewers were unanimous in leaning towards acceptance after a brief rebuttal period. I concur with their assessment, and would recommend that the paper be published in its current form.